# STRUCTURE- AND APPEARANCE-RICH TRAINING-FREE SPATIAL CONTROL FOR TEXT-TO-IMAGE GENERATION

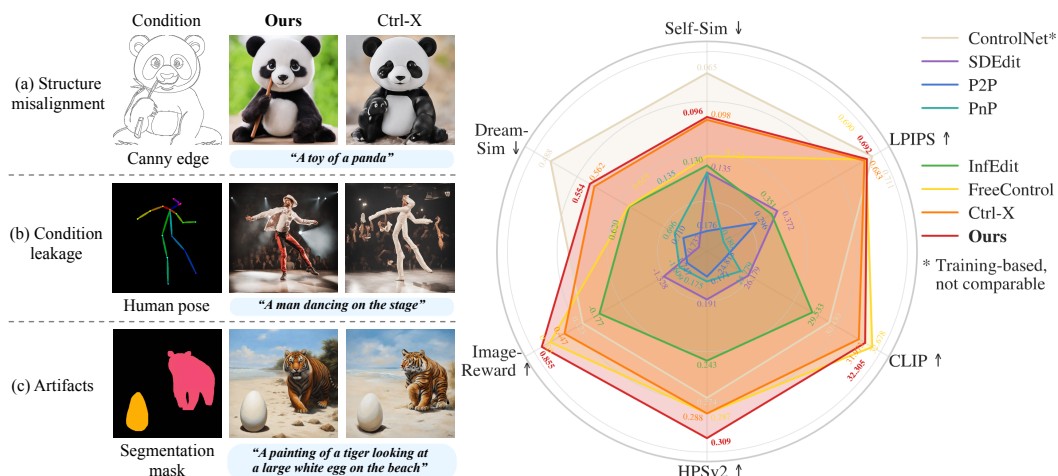

Figure 1: **Structure- and appearance-rich training-free spatial control for text-to-image generation.** We propose a training-free framework that enables high-quality spatial control for pretrained text-to-image diffusion models under arbitrary spatial conditions. (**Left**) By introducing strong and intuitive structure and appearance control, our method effectively addresses key limitations of prior work such as Ctrl-X (Lin et al., 2024), including structure misalignment, condition leakage, and artifacts, and (**Right**) achieves SOTA performance among all training-free methods; in the radar chart, greater distance from the center indicates superior results.

## ABSTRACT

Text-to-image (T2I) diffusion models have shown remarkable success in generating high-quality images from text prompts. Recent efforts extend these models to incorporate conditional images (e.g., canny edge) for fine-grained spatial control. Among them, feature injection methods have emerged as a training-free alternative to traditional fine-tuning-based approaches. However, they often suffer from structural misalignment, condition leakage, and visual artifacts, especially when the condition image diverges significantly from natural RGB distributions. Through an empirical analysis of existing methods, we identify a key limitation: the sampling schedule of condition features, previously unexplored, fails to account for the evolving interplay between structure preservation and domain alignment throughout diffusion steps. Inspired by this observation, we propose a flexible training-free framework that decouples the sampling schedule of condition features from the denoising process, and systematically investigate the spectrum of feature injection schedules for a higher-quality structure guidance in the feature space. Specifically, we find that condition features sampled from a single timestep are sufficient, yielding a simple yet efficient schedule that balances structure alignment and appearance quality. We further enhance the sampling process by introducing a restart refinement schedule, and improve the visual quality with an appearance-rich prompting strategy. Together, these designs enable training-free generation that is both structure-rich and appearance-rich. Extensive experiments show that our approach achieves state-of-the-art results across diverse zero-shot conditioning scenarios.

# 1 INTRODUCTION

With the success of text-to-image (T2I) diffusion models (Saharia et al., 2022b; Rombach et al., 2022; Podell et al., 2024), recent research has explored integrating conditional images, *e.g.*, depth maps for spatial control. Early approaches, such as ControlNet (Zhang et al., 2023), rely on fine-tuning or auxiliary networks trained on paired data, which constrains their flexibility and scalability. More recent studies have shown that the rich structural information encoded within diffusion features can be exploited to guide image generation without retraining, thereby enabling zero-shot control. They either introduce additional guidance terms to minimize the feature distance between the condition and target during denoising (Epstein et al., 2023; Bansal et al., 2023; Mo et al., 2024), or inject features extracted from the condition image at each timestep into the target image (Hertz et al., 2023; Tumanyan et al., 2023; Lin et al., 2024). Among them, feature injection-based methods such as Ctrl-X (Lin et al., 2024) have shown promising performance across diverse conditioning scenarios.

However, these methods still encounter several failures, including structural misalignment, condition leakage, and visual artifacts (Fig. 1). These issues become more pronounced when the condition image deviates significantly from natural RGB distributions, *e.g.*, in pose or depth maps (Fig. 5). This suggests that a key challenge lies in the domain gap between condition and natural image features in pretrained T2I diffusion models. We hypothesize that the injected condition features often lie outside the distribution of natural image features, which hinders the synthesis of high-fidelity results. This motivates us to analyze the temporal dynamics of diffusion features, observing a trade-off between structural fidelity and domain alignment (see Figs. 2 and 3). These findings expose a fundamental limitation in existing training-free methods (Hertz et al., 2023; Tumanyan et al., 2023; Lin et al., 2024), which rely on condition features extracted at the *same* timestep during denoising. This schedule fails to accommodate the evolving trade-off across timesteps: early features leads to loss of structural detail, while late features result in domain mismatch and condition leakage (Fig. 3).

To address this, we generalize the sampling process of condition features and explore the design space of the feature injection schedule. The result shows that the optimal timestep is neither the same one as the target output image nor the latest one with the clearest features. Through a comprehensive investigation, we identify a family of candidate schedules that share an identical last timestep, among which a constant schedule yields consistently strong results. Building on these insights, we propose a more flexible feature injection framework that decouples the injection timestep from the denoising process. To further enhance control precision and visual fidelity, we apply a restart refinement schedule that iteratively mitigates visual artifacts introduced by injected features, and incorporate prompt augmentation to ensure semantic alignment with the condition image. Together, these designs enable structure- and appearance-richer control of pretrained diffusion models (Podell et al., 2024). Fig. 4 provides an overview of our framework, which consists of three key components: (i) *Structure-Rich Injection (SRI)* injects condition features based on a principled sampling schedule; (ii) *Restart Refinement (RR)* performs iterative forward–backward denoising; (iii) *Appearance-Rich Prompting (ARP)* aligns the semantics of the appearance prompt with the condition image.

Extensive experiments validate the effectiveness of our approach across diverse types of condition images, demonstrating improved structural consistency, visual fidelity, and semantic alignment compared to state-of-the-art training-free methods. Furthermore, the idea of our framework can be readily incorporated into other training-free methods, such as FreeControl (Mo et al., 2024), where it yields notable improvements, highlighting its versatility. In summary, our contributions are threefold: (i) We reveal the inherent limitation of existing training-free methods in sampling schedules, and identify a spectrum of alternatives through a principled analysis of the design space. (ii) Building on this insight, we propose a novel framework that enables structure- and appearance-rich controllable T2I generation. (iii) Our proposed method demonstrates state-of-the-art performance in comparison to previous training-based and training-free baselines, delivering superior structure preservation, text-image alignment, and visual fidelity.

# 2 RELATED WORK

## 2.1 T2I DIFFUSION MODELS

Text-to-image (T2I) diffusion models (Ho et al., 2022; Saharia et al., 2022b; Ramesh et al., 2022; Rombach et al., 2022) typically leverage U-Net (Ronneberger et al., 2015) or transformer-based

backbones (Peebles & Xie, 2023; Esser et al., 2024; Labs, 2024) and integrate textual information via cross-attention or classifier-free guidance (Rombach et al., 2022; Ho & Salimans, 2021; Nichol et al., 2022). Latent diffusion models like Stable Diffusion (Rombach et al., 2022) introduce compressed latent spaces to reduce computational cost. In addition to architectural innovations, some work focuses on improving sampling efficiency (Song et al., 2021; Lu et al., 2022; Karras et al., 2022; Xu et al., 2023b; Liu et al., 2023; Song et al., 2023; Zhao et al., 2023b). Restart Sampling (Xu et al., 2023b) proposes alternating between adding noise and denoising to balance discretization error and contraction. Our work explores sampling strategies in the context of conditional text-to-image generation.

## 2.2 TRAINING-BASED CONTROLLABLE DIFFUSION MODELS

It is difficult to convey human preferences through text descriptions alone. Training-based controllable diffusion models mitigate this problem by training auxiliary modules or fine-tuning the model to incorporate additional input signals to guide the generation process. According to task characteristics, these methods can be broadly classified into three categories: **(i) Image editing** (Brooks et al., 2023; Goel et al., 2024; Kim et al., 2022; Wang et al., 2023; Sheynin et al., 2024; Geng et al., 2024; Xiao et al., 2025; Wu et al., 2025; Chen et al., 2025; Le et al., 2025; Xia et al., 2025; Xie et al., 2025; Han et al., 2024) takes an input image and applies targeted modifications while preserving other regions of the image; **(ii) Image-to-image translation** (Isola et al., 2017; Saharia et al., 2022a; Tumanyan et al., 2022; Ouyang et al., 2025; Park et al., 2019) learns mappings between images of different domains; **(iii) Conditional text-to-image (T2I) generation** methods synthesize images that satisfy both a text prompt and a control condition. Among these approaches, some works condition the generation on layout cues (*e.g.*, bounding boxes) (Li et al., 2023b; Yang et al., 2023; Wang et al., 2024) or reference images of specific subjects (Gal et al., 2023; Ruiz et al., 2023; 2024; Avrahami et al., 2023a; Po et al., 2024; Li et al., 2023a; Zhang et al., 2024b; 2025b; Tan et al., 2025a;b; Chen et al., 2025; Xia et al., 2025; Xiao et al., 2025; Wu et al., 2025; Le et al., 2025). Another line of work (Zhang et al., 2023; Mou et al., 2024; Ye et al., 2023; Zhao et al., 2023a; Avrahami et al., 2023b; Zhang et al., 2024b; 2025b; Tan et al., 2025a;b; Li et al., 2024; Xiao et al., 2025; Wu et al., 2025; Chen et al., 2025; Le et al., 2025; Xia et al., 2025; Xie et al., 2025; Han et al., 2024; Xu et al., 2025b; Zhao et al., 2025) enables fine-grained structural control by leveraging condition images of different modalities (*e.g.*, canny edges, OpenPose keypoints (Cao et al., 2019)). Despite their impressive performance, these methods all require retraining or fine-tuning on datasets tailored to the control signal, which limits their generalization to new model checkpoints and novel control conditions.

## 2.3 TRAINING-FREE CONTROLLABLE DIFFUSION MODELS

On the other hand, training-free controllable diffusion models operate at inference time to achieve condition control without additional training on task-specific paired data. Similar to Sec. 2.2, we categorize them into 3 groups: **(i) Image editing** (Cao et al., 2023; Xu et al., 2024b; Epstein et al., 2023; Parmar et al., 2023; Zhang et al., 2024a; Tewel et al., 2025; Jia et al., 2025; Couairon et al., 2023; Feng et al., 2025b; Dalva et al., 2024; Zhu et al., 2025; Avrahami et al., 2025; Wang et al., 2025a; Xu et al., 2025a; 2024a; Wei et al., 2025; Titov et al., 2024; Hu et al., 2025); **(ii) Image-to-image translation** (Su et al., 2023), with some works (Alaluf et al., 2024; Lin et al., 2024; Kwon & Ye, 2023; Huang et al., 2025; Go et al., 2024; Chung et al., 2024) further conditioning the transformation on an additional appearance image; **(iii) Conditional text-to-image (T2I) generation**, which generates images consistent with both textual prompts and input control signals. These signals range from coarse layout constraint such as bounding boxes (Xiao et al., 2024; Chen et al., 2024; Xie et al., 2023; Li et al., 2025; Wang et al., 2025c), semantic references like subject images (Zhang et al., 2025a; Feng et al., 2025a; Wang et al., 2025b; Ding et al., 2024; Pham et al., 2024; Rout et al., 2025), to condition images that provide fine-grained control (Lin et al., 2024; Mo et al., 2024; Tumanyan et al., 2023; Hertz et al., 2023; Bansal et al., 2023; Meng et al., 2022; Kim et al., 2023b; Lee et al., 2025). These approaches enable flexible control over pre-trained diffusion models and can generalize to novel control modalities without the cost of additional data collection or retraining. As a **training-free conditional T2I method** conditioned on **condition images**, our approach extends this line of work by improving both control fidelity and generation quality across diverse visual conditions.

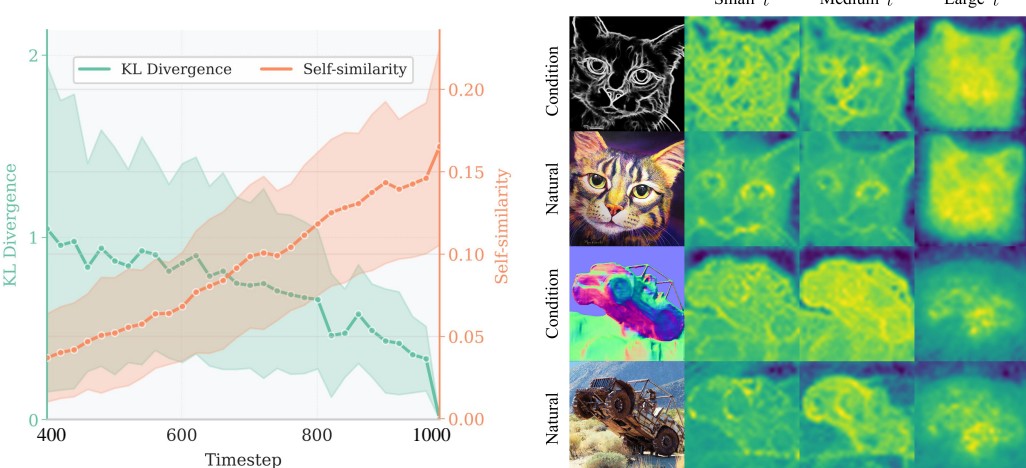

Figure 2: **The evolving curves of KL divergence and L2 distance of self-similarity matrices across diffusion timesteps**. The denoising process starts from timestep 1000 (right).

Figure 3: **Visualizing diffusion features extracted from the condition and natural images at different timesteps**. We display the first principal component computed for each time step across all images.

## 3 REVISITING SAMPLING SCHEDULES OF CONDITION FEATURES

**Background.** Given a text prompt $\mathcal{P}$ and a condition image $\mathbf{I}^{\text{struct}}$ of arbitrary modality, the goal is to generate an output image $\mathbf{I}$ that semantically aligns with the prompt $\mathcal{P}$ while preserving the structure of $\mathbf{I}^{\text{struct}}$. To align the structure of the generated image with that of the condition $\mathbf{I}^{\text{struct}}$, recent training-free approaches such as Ctrl-X (Lin et al., 2024) leverage the diffusion features of a noisy latent $\mathbf{x}_t^{\text{struct}}$ of the condition image. Specifically, they obtain a clean latent of the $\mathbf{x}_0^{\text{struct}}$ by encoding $\mathbf{I}^{\text{struct}}$ using a Variational Auto-Encoder, and then obtain its noisy version $\mathbf{x}_t^{\text{struct}}$ through DDIM inversion (Song et al., 2021) or the diffusion forward process. Intermediate features are subsequently extracted from designated layers of the model backbone, and condition features are injected into those of $\mathbf{I}$ at each timestep. We denote these condition features as $\mathbf{f}_{l,t}^{\text{struct}}$, where $l$ refers to the layer index and $t$ denotes the timestep.

**Limitations of Existing Methods.** While enabling zero-shot spatial control with diverse condition modalities, these methods often suffer from structural misalignment and condition leakage. For instance, Ctrl-X (Lin et al., 2024) fails to preserve the structure of the panda (Fig. 1). Empirically, we observe that these failures are further exacerbated when the condition image deviates substantially from natural RGB images, as in the case of pose or depth maps shown in Fig. 5. This suggests that a key challenge lies in the domain gap between the condition and natural image distributions in the feature space of pretrained diffusion models. We hypothesize that the injected features $\mathbf{f}_{l,t}^{\text{struct}}$ fall outside the distribution of natural image features, thereby reducing their effectiveness in preserving the structure of the condition image $\mathbf{I}^{\text{struct}}$ during generation.

**Empirical Analysis.** To validate this hypothesis, we quantitatively analyze features from 100 pairs of condition images across five common modalities (see Appx. C). As shown by the orange curve in Fig. 2, self-similarity distance decreases as noise is reduced, reflecting a progressive gain of fine-grained spatial cues. However, this improved structural fidelity comes at the cost of reduced domain alignment: the green curve in Fig. 2 shows that the KL divergence increases at lower timesteps, indicating a widening domain gap between natural and condition features.

We further conduct principal component analysis and visualize the diffusion features in Fig. 3. There exists a visible discrepancy between the features of the condition image and its natural counterpart, which is more pronounced at smaller timesteps. Another notable pattern is that the primary structural information intended to be preserved emerges in the middle stage, while modality-specific details become more prominent in the late stage. These observations highlight the limitations of previous methods: as the sampling schedule progresses, the early features convey only coarse structural cues,

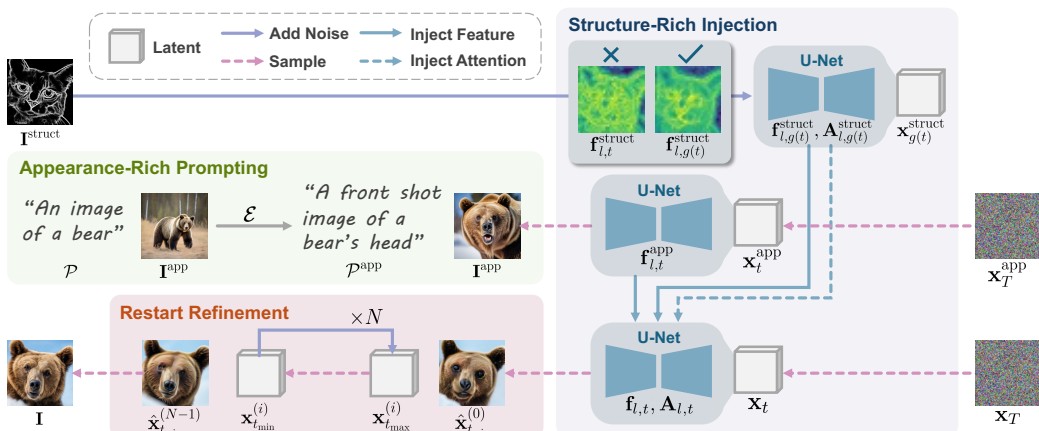

Figure 4: **Method overview.** Given a condition image $\mathbf{I}^{\text{struct}}$ and a prompt $\mathcal{P}$, our method generates an output image $\mathbf{I}$, aligning semantically with $\mathcal{P}$ while preserving the structure of $\mathbf{I}^{\text{struct}}$. Our framework consists of three key components. (i) The **Structure-Rich Injection (SRI) module** (blue) injects structure-rich condition features $\mathbf{f}^{\text{struct}}_{l,g(t)}$ and attentions $\mathbf{A}^{\text{struct}}_{l,g(t)}$ into the output feature space to enable spatial control (Sec. 4.1). (ii) The **Restart Refinement (RR) module** (pink) iteratively adds noise to and denoises $\mathbf{I}$ to refine visual details such as the eyes of the bear (Sec. 4.2). (iii) The **Appearance-Rich Prompting (ARP) module** (green) derives an enriched prompt $\mathcal{P}^{\text{app}}$ based on the semantics of the condition image $\mathbf{I}^{\text{struct}}$ to generate a reference image $\mathbf{I}^{\text{app}}$ for appearance transfer (Sec. 4.3).

whereas late features introduce out-of-distribution details in the output image, leading to structure misalignment and condition leakage.

**Our Insight.** Motivated by these observations, we generalize the sampling schedule of condition features, decoupling it from the denoising process, and explore the function space of feature injection schedules. In this framework, the structure latent from which the injected feature is extracted lies at timestep $g(t)$, with $g(t) = t$ covering the case in previous approaches as a special instance. We systematically explore different forms of $g(t)$ and evaluate their impact on structure alignment and visual quality (see Sec. 5.3 for details).

After a principled investigation, we conclude that (i) the optimal condition timestep is neither the output timestep nor the smallest (clearest) one, but lies in the middle stage; and (ii) schedules with their last timestep anchored around medium timesteps consistently deliver the optimal balance between structural fidelity and visual quality, largely independent of their functional form.

## 4 METHOD

Building on the insights in Sec. 3, we introduce a training-free controllable T2I generation framework that enables flexible, structure- and appearance-richer control.

Our approach comprises three key components, as illustrated in Fig. 4: (i) **Structure-Rich Injection (SRI)** injects condition features based on a principled sampling schedule, with a better balance of structure preservation and domain alignment (Sec. 4.1); (ii) **Restart Refinement (RR)** schedule performs iterative refinement to suppress visual artifacts and improve overall image fidelity (Sec. 4.2). (iii) **Appearance-Rich Prompting (ARP)** enriches the original prompt $\mathcal{P}$ with detailed descriptions informed by $\mathbf{I}^{\text{struct}}$, facilitating appearance guidance (Sec. 4.3). Together, these modules enable structure- and appearance-aware generation across diverse conditions, all in a zero-shot, training-free manner. We now describe each component in detail.

### 4.1 STRUCTURE-RICH INJECTION

The structure-rich injection strategy adopts a sampling schedule where the extracted condition features are both semantically compatible and structurally informative. Specifically, we begin by encoding the structure condition image $\mathbf{I}^{\text{struct}}$ using the model backbone to obtain the condition features and attention maps, as illustrated in Fig. 4. Prior work (Hertz et al., 2023; Tumanyan et al., 2023; Mo

et al., 2024; Lin et al., 2024) typically extracts condition features $\mathbf{f}_{l,t}^{\text{struct}}$ and attention maps $\mathbf{A}_{l,t}^{\text{struct}}$ at the same timestep $t$ as the denoising step used for generating the output image $\mathbf{I}$ (the bottom branch of the U-Net in Fig. 4), where $l$ denotes the specific U-Net layer to be injected.

According to the empirical analysis in Sec. 3, however, we select features from a separate schedule $g(t)$, where $g(\cdot)$ is a general function of the current timestep $t$. As shown in the blue block of Fig. 4, the extracted features $\mathbf{f}_{l,g(t)}^{\text{struct}}$ and attention maps $\mathbf{A}_{l,g(t)}^{\text{struct}}$ are then used to replace their counterparts $\mathbf{f}_{l,t}$ and $\mathbf{A}_{l,t}$ in the generation backbone at timestep $t$:

$$\mathbf{f}_{l,t} \leftarrow \mathbf{f}_{l,g(t)}^{\text{struct}} \quad \text{and} \quad \mathbf{A}_{l,t} \leftarrow \mathbf{A}_{l,g(t)}^{\text{struct}}. \tag{1}$$

Note that we only apply our structure-rich injection for timesteps $t \geq \tau$, where $\tau$ denotes the structure control schedule.

**Single-Step Sampling and Caching.** Empirical findings in Sec. 3 demonstrate that medium-timestep schedules consistently yield the optimal balance between structure preservation and visual quality, largely independent of function forms, and even a constant-timestep schedule achieves competitive results. Guided by this observation, we adopt a constant schedule $g(t) = 600$ for all subsequent experiments. An additional benefit of a constant schedule is computational efficiency. Since the features of the condition image need to be computed only once, they can then be cached and reused throughout the denoising process. Please refer to Sec. 5 and Appx. F.1 for detailed results.

### 4.2 RESTART REFINEMENT

Despite a carefully designed injection schedule, structure-rich features can still introduce out-of-distribution artifacts and condition leakage during denoising. To address this, we adopt a restart refinement schedule inspired by diffusion-based sampling methods (Xu et al., 2023b). As illustrated in the pink block of Fig. 4, after several rounds of structure and appearance control, we inject noise at an intermediate timestep $t_{\min}$, effectively restarting the denoising process by transitioning the latent to $t_{\max}$ step. A DDIM backward step (Song et al., 2021) is then applied. This forward–backward cycle is repeated $N$ times within $[t_{\min}, t_{\max}]$. In the $i^{\text{th}}$ iteration ($i \in \{0, \dots, N-1\}$), the restart proceeds as follows:

$$(\text{Forward}) \ \mathbf{x}_{t_{\max}}^{(i+1)} = \mathbf{x}_{t_{\min}}^{(i)} + \epsilon_{t_{\min} \to t_{\max}}, \quad (\text{Backward}) \ \mathbf{x}_{t_{\min}}^{(i+1)} = h_{\text{DDIM}}(\mathbf{x}_{t_{\max}}^{(i+1)}, t_{\max} \to t_{\min}),$$

where the initial $\mathbf{x}_{t_{\min}}^{(0)}$ is obtained by simulating the DDIM step until $t_{\min}$: $\mathbf{x}_{t_{\min}}^{(0)} = h_{\text{DDIM}}(\mathbf{x}_T, T \to t_{\min})$, and the noise $\epsilon_{t_{\min} \to t_{\max}}$ is sampled from the perturbation kernel from $t_{\min}$ to $t_{\max}$. Through this schedule, our approach achieves better visual fidelity, as demonstrated in the ablation study in Fig. 8.

### 4.3 APPEARANCE-RICH PROMPTING

To enhance the realism of the generated image, prior work (Mo et al., 2024; Lin et al., 2024) generates an auxiliary appearance image $\mathbf{I}^{\text{app}}$ and perform appearance transfer (the middle branch of the U-Net in Fig. 4; see Appx. D for technical details). However, brief and ambiguous user prompts sometimes hinder the establishment of semantic correspondence between the appearance image $\mathbf{I}^{\text{app}}$ and the output image $\mathbf{I}$ in existing appearance transfer methods, leading to artifacts. For example, as illustrated in Fig. 4, the condition image is a frontal view of a cat's head, while the text prompt specifies "a bear", resulting in an appearance image $\mathbf{I}^{\text{app}}$ that depicts a full-body bear.

To tackle this issue, we propose Appearance-Rich Prompting (ARP), a strategy that leverages multimodal large language models (Achiam et al., 2023) to systematically align the semantics of $\mathcal{P}$ with those of the conditions $\mathbf{I}^{\text{struct}}$, as shown in the green block of Fig. 4. Illustrative examples are provided in the ablation study in Fig. 8. Please refer to Appx. D for details of the prompt engineering pipeline $\mathcal{E}$.

## 5 EXPERIMENTS

### 5.1 SETUP

**Dataset.** We base our evaluation on datasets from prior work (Mo et al., 2024; Lin et al., 2024). However, many of the condition types in prior datasets are underrepresented, resulting in an imbalanced distribution. To enable more consistent evaluation, we collect seven commonly used structural

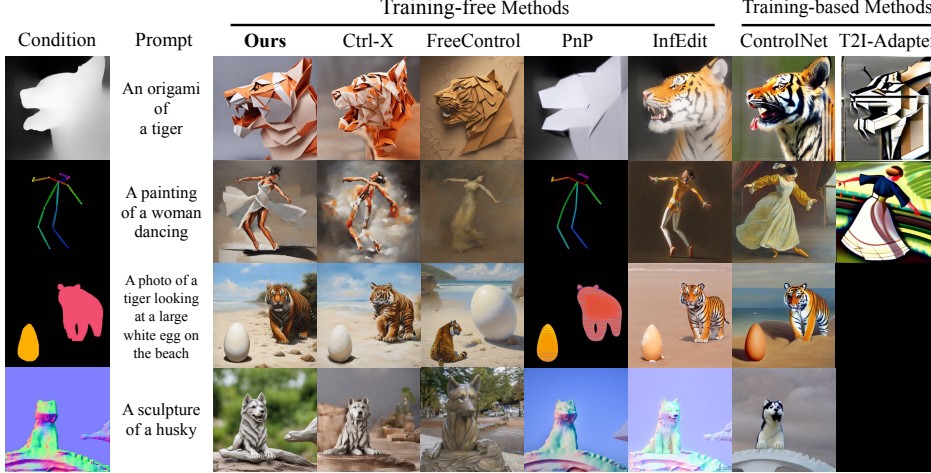

Figure 5: **Qualitative comparison with existing methods**. Our method effectively addresses common failure modes observed in previous methods: structure misalignment, condition leakage, and visual artifacts, generating high-quality images that adhere closely to the prompts with strong spatial alignment.

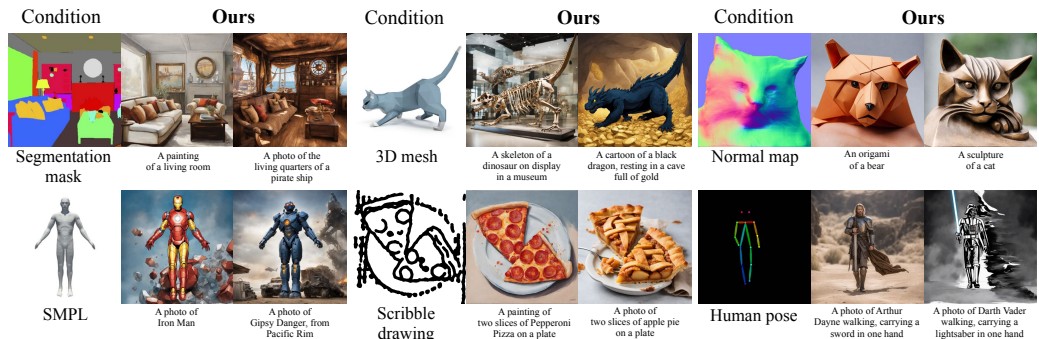

Figure 6: **Qualitative results for more control conditions**. Our method can handle a variety of challenging condition images and prompts, including those infeasible for training-based approaches.

conditions: canny edges, depth maps, normal maps, human poses, segmentation masks, HED edges, and scribble drawings, and construct a more balanced dataset with over 20 image-text pairs for each condition type.

**Baselines.** We evaluate our method against 6 existing training-free baselines: Ctrl-X (Lin et al., 2024), FreeControl (Mo et al., 2024), PnP (Tumanyan et al., 2023), P2P (Hertz et al., 2023), SDEdit (Meng et al., 2022), and InfEdit (Xu et al., 2024a), as well as 2 training-based baselines: ControlNet (Zhang et al., 2023) and T2I-Adapter (Mou et al., 2024). Experiment results on four condition types supported by T2I-Adapter-SDXL (canny, depth, normal, and pose) are provided in Appx. F.3. Wherever possible, we implement each method using SDXL 1.0 (Podell et al., 2024); otherwise, we use their best-performing publicly available checkpoints. To ensure a fair comparison, we use 50 denoising steps and 50 inversion steps for all baselines.

**Evaluation Metrics.** Following prior work (Mo et al., 2024; Lin et al., 2024), we employ three widely-adopted metrics for comparison. (i) CLIP score (Radford et al., 2021) measures text-image alignment via cosine similarity between the CLIP embeddings of the generated image and text prompt. (ii) DINO self-similarity score (Self-sim) (Caron et al., 2021; Tumanyan et al., 2022) quantifies structural alignment in the DINO-ViT feature space. (iii) Condition LPIPS score (LPIPS) (Zhang et al., 2018) measures perceptual deviation between the generated image and the condition image.

In addition to traditional metrics, we further adopt three reward model metrics that more accurately reflect human preferences. (i) DreamSim (Fu et al., 2023) evaluates the perceptual similarity of two

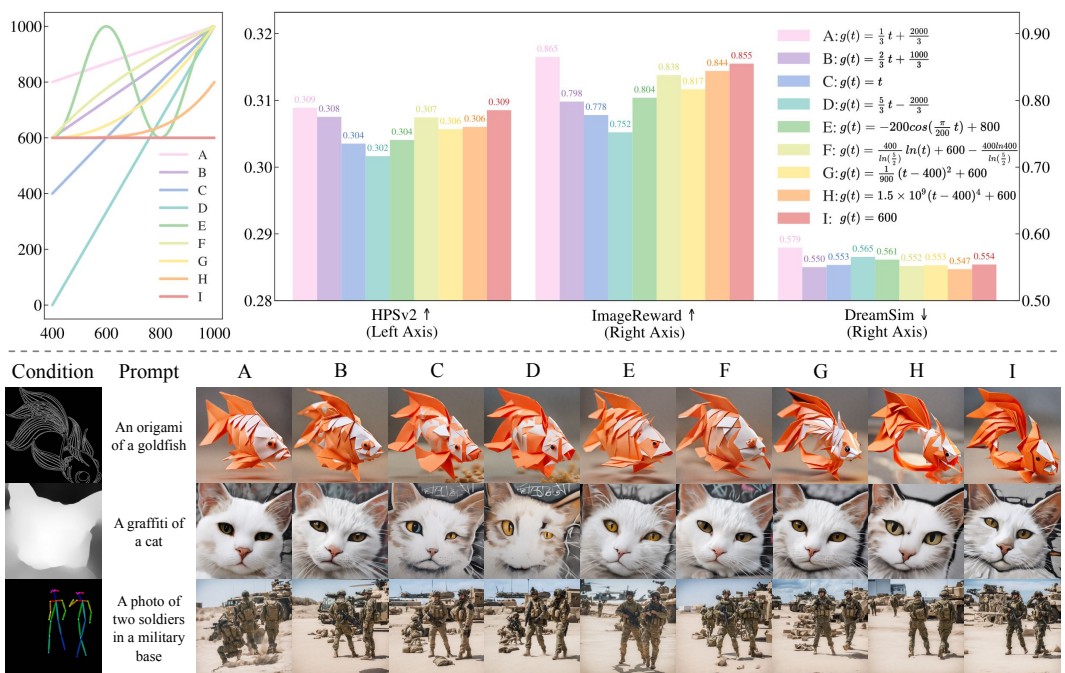

Figure 7: **Ablation of different injection schedules of SRI**. We report quantitative (***Top***) and qualitative (***Bottom***) results for different injection schedules.

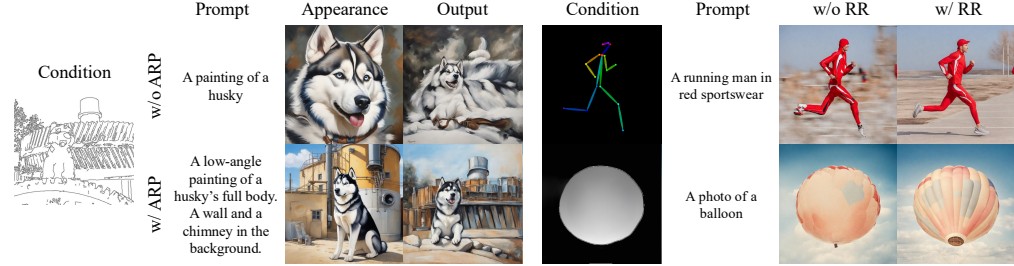

Figure 8: **Ablation of ARP and RR**. (***Left***) ARP mitigates incorrect appearance transfers and reduces artifacts. (***Right***) RR significantly reduces condition leakage and appearance artifacts while maintaining structural alignment. See Fig. 15 and Fig. 16 in the appendix for more cases.

images by capturing both mid-level similarities (image layout, object pose, semantic content) and low-level attributes (color, texture). (ii) ImageReward (Xu et al., 2023a) measures image quality and text-image alignment based on a reward model trained with human feedback. (iii) HPSv2 (Wu et al., 2023) serves as a reliable indicator of overall generation quality aligned with human judgments. To ensure the accuracy of the results, all quantitative comparisons and ablation studies were repeated three times, and we report the mean results across these runs.

## 5.2 COMPARISON WITH STATE-OF-THE-ART (SOTA)

**Analysis.** Figs. 1 and 5 present quantitative and qualitative comparisons between our method and existing baselines, respectively. While training-based approaches like ControlNet (Zhang et al., 2023) and T2I-Adapter (Mou et al., 2024) exhibit lower Self-sim scores, they often fail to adhere to the text prompts (*e.g.*, *origami*, *white egg*, *sculpture* in Fig. 5), leading to impaired text-image alignment. In contrast, our method achieves robust structural alignment while maintaining superior text-image consistency.

On the other hand, training-free baselines exhibit several limitations. SDEdit (Meng et al., 2022), PnP (Tumanyan et al., 2023), and P2P (Hertz et al., 2023) are prone to condition leakage, producing outputs that closely resemble the condition image. FreeControl (Mo et al., 2024) and InfEdit (Xu

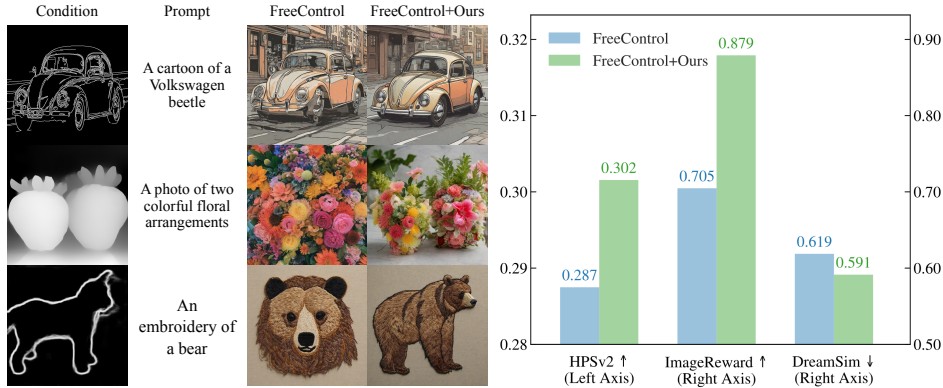

Figure 9: **Integrating our method as a plug-in into FreeControl (Mo et al., 2024)**. Our method consistently improves FreeControl on both quantitative and qualitative evaluations, achieving stronger structure fidelity and perceptual quality.

|  | Time (s) | Memory (GB) | Preference Rate |
|---|---|---|---|
| InfEdit (Xu et al., 2024a) | 31.84 | 52.44 | 11.42% |
| FreeControl (Mo et al., 2024) | 781.57 | 55.46 | 10.67% |
| Ctrl-X (Lin et al., 2024) | 19.37 | 18.79 | 21.67% |
| **Ours** | **18.79** | **18.77** | **56.25%** |

Table 1: **Computational efficiency and user study.** Our method achieves the fastest inference speed (18.79s per image) among strong baselines including Ctrl-X (Lin et al., 2024) and FreeControl (Mo et al., 2024). Moreover, 56.25% of human users prefer our method over the baselines.

et al., 2024a) yield unstable results, often generating images with inferior text-image alignment and artifacts (rows 1-4). Ctrl-X (Lin et al., 2024) performs reliably in many cases but still suffers structural misalignment (row 4), condition leakage (row 2), and artifacts (rows 1 and 3) as shown in Fig. 5. In contrast, our method consistently outperforms these baselines in structural preservation, text-image alignment, and visual quality, excelling in difficult scenarios such as abstract conditions (*e.g.*, pose), multi-object scenes, and challenging prompts (*e.g.*, *origami*). Quantitative evaluations further confirm the advantage of our approach. As shown in Fig. 1, our method surpasses training-free baselines across nearly all metrics. Please refer to Figs. 19 and 20 and Tab. 4 in the appendix for additional results.

Similar to prior studies (Fu et al., 2023; Xu et al., 2023a; Wu et al., 2023; Ma et al., 2025), we find that reward models exhibit stronger alignment with human preferences than traditional metrics. Accordingly, in the subsequent experiments, we primarily report results based on three reward-model metrics.

**User Study.** We further conduct human evaluations to validate the effectiveness of our framework. We compare our method against 3 strongest baselines: InfEdit (Xu et al., 2024a), FreeControl (Mo et al., 2024) and Ctrl-X (Lin et al., 2024), and randomly sample 30 cases from the dataset. 40 participants with related backgrounds are asked to select the most preferred result for each case, accounting for structure alignment with the condition image, semantic consistency with the prompt, and visual quality. Tab. 1 shows the result of the human evaluation, where 56.25% of participants prefer the results produced by our method. Again, this highlights the effectiveness of our approach. Please refer to Appx. E for more details of experiment execution.

**Computational Efficiency.** While the RR module introduces more sampling steps, our single-step sampling and caching strategy effectively eliminate redundant feature computations, ensuring high efficiency. Tab. 1 reports the average inference time and memory used by the 4 strongest methods on a single A800 GPU. Our method achieves the fastest inference speed and the relatively low memory cost, confirming the computational efficiency of our method. Please refer to Appx. F for the time consumption of each module within our method.

## 5.3 Ablation Study

**Structure-Rich Injection.** In this section, we explore different forms of injection schedules $g(t)$ and evaluate their impact on structure alignment and visual fidelity. The results are shown in Fig. 7. Our ablation on linear schedules (Fig. 7, A–D) reveals a clear trade-off: schedules biased toward larger timesteps (A) weaken structural alignment and increase DreamSim scores, whereas those biased toward smaller timesteps (C–D) degrade both visual quality (lower HPSv2 and ImageReward scores) and structural fidelity due to excessive modality-specific cues. In contrast, the medium-timestep schedules (B) strike a favorable balance, yielding stronger structure preservation without sacrificing visual realism.

Beyond linear functions, we further examine a family of schedules (E–I) that share the same final timestep but differ in monotonicity, convexity, and initialization. Interestingly, all of them deliver consistently strong performance, suggesting robustness to functional variations. Empirically, we find that convex functions slightly outperform concave ones, and even a constant-timestep schedule achieves competitive results.

**Appearance-Rich Prompting.** Fig. 8 demonstrates that the ARP module effectively adapts the prompt to capture key visual attributes of the condition image. As shown in Tab. 2, removing the ARP module decreases performance across all three metrics, verifying its effectiveness in enhancing structural preservation and visual fidelity.

|  | Dream-Sim ↓ | Image-Reward ↑ | HPSv2 ↑ |
|---|---|---|---|
| w/o ARP | 0.558 | 0.799 | 0.308 |
| w/o RR | 0.544 | 0.518 | 0.286 |
| **Ours** | 0.554 | 0.855 | 0.309 |

Table 2: **Quantitative ablation of ARP and RR.**

**Restart Refinement.** As shown in Fig. 8, the RR schedule mitigates both condition leakage and artifacts, leading to improved generation quality while maintaining strong structural alignment. Tab. 2 suggests that RR improves visual fidelity but partly compromises structural alignment, which is consistent with its intended design: it relaxes overly rigid structural constraints in order to mitigate condition leakage (see Fig. 1) and improve perceptual quality.

## 5.4 Improvement over Prior Methods

Our pipeline can serve as a plug-in to enhance other U-Net-based conditional T2I approaches. As an example, we applied our method to FreeControl (Mo et al., 2024), another recent conditional T2I model. As shown in Fig. 9, our approach consistently improves FreeControl across three evaluation metrics. Qualitative comparisons in Fig. 9 further highlight that FreeControl with our method yields outputs with both stronger structural preservation and higher perceptual quality.

## 6 Conclusion

We propose a training-free framework for conditional text-to-image (T2I) generation. By leveraging the features of pretrained diffusion models in a principled manner, our method balances structural fidelity and appearance quality while automatically enhancing image realism and prompt relevance. Our investigation facilitates the understanding of the feature space of T2I diffusion models and achieves a strong, general, and robust solution for injection-based pipelines.

**Limitations and future directions.** Although we draw meaningful conclusions from principled investigations and design an effective method, it remains a promising future direction to interpret the empirical results theoretically. A formal explanation in a high-dimensional feature space is a non-trivial task, and it requires further dedication from the research community. The second limitation of our approach is that the appearance-rich prompting module requires access to a multimodal language model. We recognize that it may raise concerns regarding privacy and safety, and hope our findings and analysis can shed light on controllable visual content creation.

## 7 STATEMENTS

**Ethics statement.** Although our experiments include human pose conditions, we do not involve human faces or any personally identifiable data, so there are no direct privacy concerns. Our method enables higher-quality and more realistic controllable generation without additional training or optimization. However, this capability and accessibility also raise the risk of malicious use of pretrained generative models (e.g., deepfakes). We urge the community not to misuse our method for deceptive or harmful purposes, such as spreading misinformation or generating non-consensual content. In response to such safety concerns, large generative models have increasingly incorporated safeguards. Likewise, our framework can inherit these protections, as it is built on a pretrained backbone, and its plug-and-play nature allows the open-source community to scrutinize and enhance its safety.

**Reproducibility statement.** We include full implementation details and experimental setups in the appendix to ensure reproducibility. For a complete explanation of our analysis of diffusion features, please refer to Appx. C. We provide the implementation details of our proposed method as well as a complete description of the experimental setups in Appx. D and Appx. E.

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

APPENDIX

We first provide LLM usage statement in Appx. A. We provide preliminaries in Appx. B. In Appx. C, we further analyze the domain gap and structure preservation of diffusion features. Then we elaborate on the implementation details of our proposed method in Appx. D and the experimental setups in Appx. E. We show additional experimental results in Appx. F.

## A   THE USE OF LARGE LANGUAGE MODELS (LLMS)

We used large language models (LLMs) to assist in refining the paper's writing and producing the appearance prompt in the ARP module. We used LLMs to enable ARP in the experiments. LLMs played no significant role in the research ideation of this paper.

## B   PRELIMINARIES

**Diffusion Models.** Diffusion models are a family of probabilistic generative models characterized by two processes.

The *forward process* iteratively adds Gaussian noise to a clean image $\mathbf{x}_0$ to obtain $\mathbf{x}_t$ for time step $t \sim [1, T]$, which can be reparameterized in terms of a noise schedule $\alpha_t$ where

$$\mathbf{x}_t = \sqrt{\alpha_t}\mathbf{x}_0 + \sqrt{1 - \alpha_t}\epsilon, \tag{2}$$

for $\epsilon \sim \mathcal{N}(0, \mathbb{I})$.

The *backward process* generates images by iteratively denoising an initial Gaussian noise $\mathbf{x}_T \sim \mathcal{N}(0, \mathbb{I})$, also known as diffusion sampling (Ho et al., 2020). This process uses a parameterized denoising network $\epsilon_\theta$ conditioned on a text prompt $\mathcal{P}$, where at time step $t$ we obtain a cleaner $\mathbf{x}_{t-1}$ as

$$\mathbf{x}_{t-1} = \sqrt{\alpha_{t-1}}\hat{\mathbf{x}}_t + \sqrt{1 - \alpha_{t-1}}\epsilon_\theta(\mathbf{x}_t \mid t, \mathcal{P}), \tag{3}$$

$$\hat{\mathbf{x}}_t = \frac{\mathbf{x}_t - \sqrt{1 - \alpha_t}\epsilon_\theta(\mathbf{x}_t \mid t, \mathcal{P})}{\sqrt{\alpha_t}}. \tag{4}$$

Intuitively, $\hat{\mathbf{x}}_t$ approximates the initial clean image, which is subsequently perturbed with an appropriate amount of noise to produce the input for the following timestep.

**Guidance.** The iterative inference of diffusion enables people to guide the sampling process on auxiliary information. *Guidance* modifies Eq. (3) to compose additional score functions that point toward richer and specifically conditioned distributions (Bansal et al., 2023; Epstein et al., 2023), expressed as

$$\hat{\epsilon}_\theta(\mathbf{x}_t \mid t, \mathcal{P}) = \epsilon(\mathbf{x}_t \mid t, \mathcal{P}) - s\,\mathbf{g}(\mathbf{x}_t \mid t, y), \tag{5}$$

where $\mathbf{g}$ is an energy function and $s$ is the guidance strength. In practice, $\mathbf{g}$ can range from classifier-free guidance (where $\mathbf{g} = \epsilon$ and $y = \emptyset$, *i.e.* the empty prompt) to improve image quality and prompt adherence for T2I diffusion (Ho & Salimans, 2021; Rombach et al., 2022), to arbitrary gradients computed from auxiliary models or diffusion features common to guidance-based controllable generation (Bansal et al., 2023; Epstein et al., 2023; Mo et al., 2024). Thus, guidance provides the customizability on the type and variety of conditioning for controllable generation, as it merely requires a differentiable loss with respect to $\mathbf{x}_t$. However, the need for backpropagation during inference often leads to increased memory consumption and slower inference speed. Moreover, guidance-based methods often fail to capture fine structural details in controllable generation tasks.

**Diffusion U-Net architecture.** Many pretrained T2I diffusion models are text-conditioned U-Nets, which contain an encoder and a decoder that downsample and then upsample the input $\mathbf{x}_t$ to predict $\epsilon$, with long skip connections between matching encoder and decoder resolutions (Ho et al., 2020; Rombach et al., 2022; Podell et al., 2024). Each encoder/decoder block contains convolution layers, self-attention layers, and cross-attention layers: The first two control both structure and appearance, and the last injects textual information. Thus, many training-free controllable generation methods utilize these layers, through direct manipulation (Hertz et al., 2023; Tumanyan et al., 2023; Kim et al., 2023a; Alaluf et al., 2024; Xu et al., 2024a) or for computing guidance losses (Epstein et al., 2023;

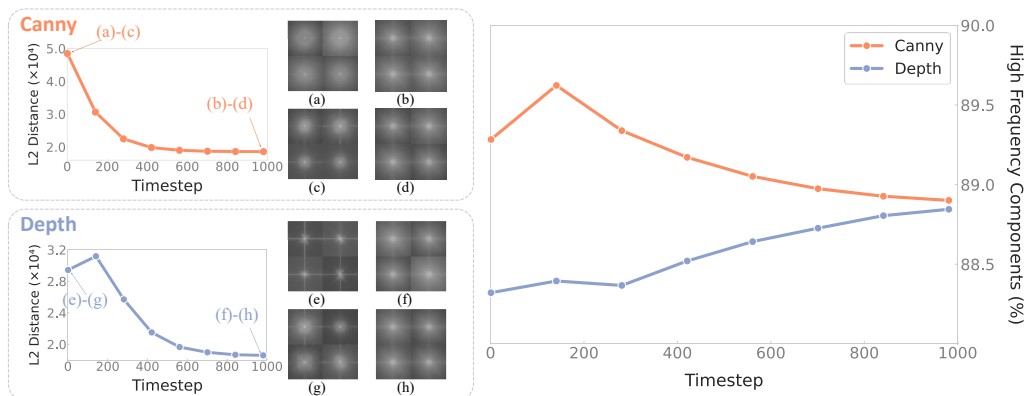

Figure 10: **Fourier analysis of noisy latents under canny edge and depth map conditions**. (*Left*) Average L2 distance between natural and condition image DFT spectra over timesteps. Subfigures (a)–(d) and (e)–(h) show the DFT spectra of four randomly selected images for both conditions at different timesteps. In each group, (a/e) and (b/f) correspond to condition latents at $t_{\text{low}}$ and $t_{\text{high}}$, while (c/g) and (d/h) correspond to natural latents at $t_{\text{low}}$ and $t_{\text{high}}$, respectively. (*Right*) Average high-frequency component ratio over timesteps.

Mo et al., 2024), with self-attention most commonly used. Let $\mathbf{f}_{l,t} \in \mathbb{R}^{HW \times c}$ be the diffusion feature with height $H$, width $W$, and channel size $c$ at time step $t$ right before attention layer $l$. Then, the self-attention operation is

$$\mathbf{Q} = \mathbf{f}_{l,t}\mathbf{W}_l^Q, \quad \mathbf{K} = \mathbf{f}_{l,t}\mathbf{W}_l^K, \quad \mathbf{V} = \mathbf{f}_{l,t}\mathbf{W}_l^V,$$

$$\mathbf{f}_{l,t} \leftarrow \mathbf{A}\mathbf{V}, \quad \mathbf{A} = \text{softmax}\left(\frac{\mathbf{Q}\mathbf{K}^\top}{\sqrt{d}}\right), \tag{6}$$

where $\mathbf{W}_l^Q, \mathbf{W}_l^K, \mathbf{W}_l^V \in \mathbb{R}^{c \times d}$ are linear transformations which produce the query $\mathbf{Q}$, key $\mathbf{K}$, and value $\mathbf{V}$, respectively, and $d$ is the dimensionality of the attention space. The $\text{softmax}$ operation is applied across the second $(HW)$-dimension (typically, $c = d$ in diffusion models). Intuitively, the attention map $\mathbf{A} \in \mathbb{R}^{(HW) \times (HW)}$ encodes how each pixel in $\mathbf{Q}$ corresponds to each in $\mathbf{K}$, which then rearranges and weighs $\mathbf{V}$. The rich structural information embedded in U-Net features lays the foundation for extensive training-free controllable generation approaches, and, together with the common issues of training-free methods, motivates us to study the temporal dynamics of diffusion features.

## C    ADDITIONAL ANALYSES

### C.1    KL DIVERGENCE

To analyze the domain gap between natural images and condition images, we collect 20 natural images from the *ImageNet-T2IR* dataset from (Tumanyan et al., 2023). Then we use the ControlNet processor (Zhang et al., 2023) to convert these natural images into 5 conditions (canny edge, depth map, normal map, HED edge, and scribble drawing), resulting in 100 natural-condition image pairs.

To quantify the distributional difference, we extract diffusion features at a fixed timestep for each image, flatten them into feature maps (size $(HW) \times F$), and concatenate all features from each domain. We then apply PCA to the combined feature set and retain only the first principal component. Each image is thus projected into a 1-dimensional vector of $HW$ values along this dominant component.

We estimate a probability distribution over these projections for each domain using Gaussian KDE. Specifically, we sample 1000 evenly spaced points between the minimum and maximum values observed in the two distributions. We then compute the KL divergence between the estimated densities of condition and natural images:

$$\text{KL}(P\|Q) = \sum_{i=1}^{1000} p(x_i) \log \frac{p(x_i)}{q(x_i)}, \tag{7}$$

where $p(x)$ and $q(x)$ denote the normalized KDE densities of condition and natural images, respectively. We repeat this computation across timesteps to observe how the domain gap evolves during the diffusion process.

## C.2 SELF-SIMILARITY

Following (Tumanyan et al., 2022), we adopt the DINO self-similarity distance Caron et al. (2021) to quantify structural similarity between images. In Vision Transformer (ViT) (Dosovitskiy et al., 2021), an image is first divided into a sequence of non-overlapping patches, which are then linearly embedded and processed as tokens. In each Transformer layer, the tokens are projected into queries, keys, and values as follows:

$$\mathbf{Q}_l = \mathbf{T}_{l-1}\mathbf{W}_l^Q, \ \mathbf{K}_l = \mathbf{T}_{l-1}\mathbf{W}_l^K, \ \mathbf{V}_l = \mathbf{T}_{l-1}\mathbf{W}_l^V, \tag{8}$$

where $\mathbf{T}_l(\mathbf{I})$ denotes the output tokens for layer $l$ for image $\mathbf{I}$, and $\mathbf{W}_l^Q$, $\mathbf{W}_l^K$, and $\mathbf{W}_l^V$ are the corresponding query, key, and value weight matrices, respectively.

To capture an image's internal structure, we compute its DINO self-similarity matrix at the final Transformer layer $L$:

$$S_L(\mathbf{I})_{ij} = \cos\_\mathrm{sim}\left(k_L(\mathbf{I})_i, k_L(\mathbf{I})_j\right), \tag{9}$$

where $\mathbf{K}_L(\mathbf{I}) = [k_L(\mathbf{I})_{cls}, k_L(\mathbf{I})_1, \ldots, k_L(\mathbf{I})_n]$ are the key embeddings from the last layer for image $\mathbf{I}$ ($n$ denotes the number of patch tokens), and $\cos\_\mathrm{sim}$ denotes cosine similarity.

As shown in (Tumanyan et al., 2022), this self-similarity-based descriptor can effectively capture the structure of an image while ignoring appearance details. Given two images $\mathbf{I}_1$ and $\mathbf{I}_2$, their structural distance is computed as the $\ell_2$ distance between their self-similarity matrices:

$$\mathcal{L}^{\mathrm{struct}} = \|S_L(\mathbf{I}_1) - S_L(\mathbf{I}_2)\|_2, \tag{10}$$

where $S_L(\mathbf{I})$ is defined in Eq. (9).

## C.3 DISCRETE FOURIER TRANSFORMATION (DFT)

As an alternative to quantifying the domain gap between natural and condition images, we employ the Discrete Fourier Transformation (DFT) to analyze differences in their frequency components. Specifically, we begin by extracting diffusion feature maps for natural and condition images at a fixed timestep, following the method described in Appx. C.1. Since DFT typically operates on spatial images rather than high-dimensional feature tensors, we use the diffusion decoder to transform these feature maps back into RGB images. We then apply DFT to the decoded images to obtain their frequency spectra and compute the L2 distance between the spectra of natural and condition image pairs.

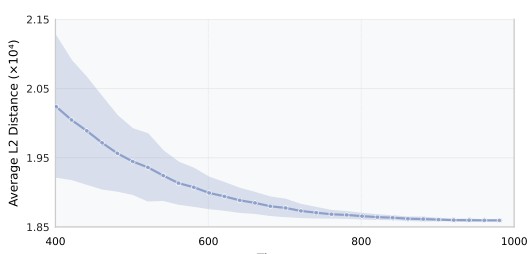

Figure 11: **Average L2 distance between natural and condition image DFT spectra over diffusion timesteps .** Results are averaged over all five conditions.

This process is repeated across all diffusion timesteps, and the resulting distances are averaged over the same 100 natural-condition image pairs as described in Appx. C.1. As shown in Fig. 11, the average L2 distance between the frequency spectra decreases progressively as the diffusion timestep increases. This trend indicates that the diffusion process gradually reduces the frequency-domain gap between natural and condition images—consistent with our findings in Sec. 3.

To further investigate how frequency components evolve through the diffusion process, we conduct a detailed analysis on two representative conditions: canny edge and depth map. Intuitively, canny edges, characterized by sharp edges and detailed contours, are expected to exhibit a higher proportion of high-frequency components in their DFT spectrum. In contrast, depth maps tend to be dominated by smooth gradients, suggesting a stronger presence of low-frequency components.

As illustrated in Fig. 10, the average L2 distance between the DFT spectra of natural and condition latents decreases over time for both conditions, consistent with the trend shown in Fig. 11. We also

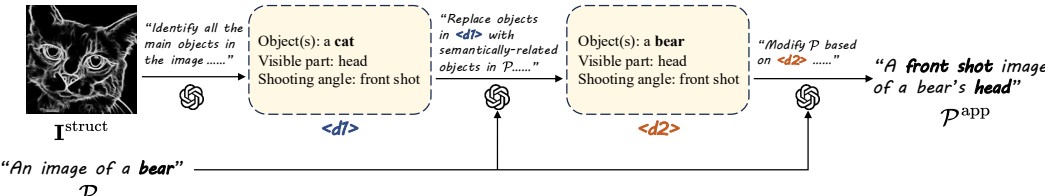

Figure 12: **Illustration of the Appearance-Rich Prompting (ARP) module**. Given the original text prompt $\mathcal{P}$, our module derives an appearance-rich prompt $\mathcal{P}^{\text{app}}$ by integrating semantic information from the condition image $\mathbf{I}^{\text{struct}}$.

visualize DFT spectra of both image types at two representative timesteps of the denoising trajectory, denoted as $t_{\text{low}}$ and $t_{\text{high}}$. In practice, we set $t_{\text{low}} = 1$ and $t_{\text{high}} = 981$. Since SDXL inference performs 50 denoising steps, steps 1 and 981 correspond to the lowest and highest noise levels in the denoising process. At $t_{\text{low}}$, canny edge spectra exhibit dispersed high-activation regions, indicative of prominent high-frequency composition. In contrast, depth map spectra show energy concentrated near the center, reflecting low-frequency dominance. Both differ markedly from the corresponding spectra of natural images. At $t_{\text{high}}$, due to accumulated noise, the DFT spectra for all images become visually similar.

This pattern is further confirmed by the right panel of Fig. 10, which plots the ratio of high-frequency components—defined as the proportion of DFT energy outside a centered circle with a radius equal to one-sixth of the image size—over timesteps. Initially, canny features are dominated by high-frequency content, while depth exhibits more low-frequency patterns. These differences gradually converge, reflecting a narrowing frequency-domain gap between different conditions.

## D    METHOD DETAILS

### D.1    SPATIALLY-AWARE APPEARANCE TRANSFER

We build our method on top of the spatially-aware appearance transfer mechanism proposed in Ctrl-X (Lin et al., 2024). Specifically, given diffusion features $\mathbf{f}_{l,t}^{\text{out}}$ and $\mathbf{f}_{l,t}^{\text{app}}$ from the output and appearance branches respectively, Ctrl-X (Lin et al., 2024) computes a cross-image attention map as follows:

$$\mathbf{A} = \text{softmax}\left(\frac{\mathbf{Q}^{\text{out}}\mathbf{K}^{\text{app}\top}}{\sqrt{d}}\right),$$

$$\mathbf{Q}^{\text{out}} = \text{norm}(\mathbf{f}_{l,t}^{\text{out}})\mathbf{W}_l^Q, \quad \mathbf{K}^{\text{app}} = \text{norm}(\mathbf{f}_{l,t}^{\text{app}})\mathbf{W}_l^K, \tag{11}$$

where $\text{norm}(\cdot)$ is applied across the spatial dimension ($HW$) and removes global statistics across spatial dimensions to isolate structural correspondence.

Subsequently, attention-weighted statistics are computed from the appearance features:

$$\mathbf{M} = \mathbf{A}\mathbf{f}_{l,t}^{\text{app}},$$

$$\mathbf{S} = \sqrt{\mathbf{A}(\mathbf{f}_{l,t}^{\text{app}} \odot \mathbf{f}_{l,t}^{\text{app}}) - (\mathbf{M} \odot \mathbf{M})}, \tag{12}$$

which are then used to modulate the output features:

$$\mathbf{f}_{l,t}^{\text{out}} \leftarrow \mathbf{S} \odot \mathbf{f}_{l,t}^{\text{out}} + \mathbf{M}. \tag{13}$$

### D.2    APPEARANCE-RICH PROMPTING

Directly using the original prompt for appearance transfer may lead to artifacts in the generated image, since such prompts tend to be brief and lacking in semantic correspondence with the condition image (see Sec. 4.3 for details). To overcome this limitation, we propose a pipeline that enriches the original text prompt $\mathcal{P}$ with semantic information extracted from the structure condition image $\mathbf{I}^{\text{struct}}$, yielding a more appearance-rich prompt $\mathcal{P}^{\text{app}}$ for generating the final appearance image $\mathbf{I}^{\text{app}}$. As illustrated in Fig. 12, we first utilize GPT-4o Achiam et al. (2023) to extract key semantic entities from the

condition image to produce dictionary $\langle d1 \rangle$. To facilitate semantic alignment between the condition image and the text prompt, we further employ GPT-4o Achiam et al. (2023) to identify and associate these extracted entities with semantically-related elements in the original text, modifying $\langle d1 \rangle$ to produce $\langle d2 \rangle$. Finally, we revise the original prompt $\mathcal{P}$ using the extracted semantic information, producing an enhanced appearance prompt $\mathcal{P}^{\text{app}}$. To help the multimodal LLM correctly follow instructions and mitigate erroneous semantic transfer, our pipeline stores intermediate information in structured dictionaries, enabling more controlled and interpretable prompt editing. More examples of the Appearance-Rich Prompting (ARP) module are provided in Fig. 15. For the full prompt used with GPT-4o Achiam et al. (2023) for Appearance-Rich Prompting, see the accompanying .txt file in the supplementary material.

# E EXPERIMENT SETUP DETAILS

## E.1 IMPLEMENTATION DETAILS

We implement our method with Diffusers (von Platen et al., 2022) on SDXL 1.0 (Podell et al., 2024) and adopt the same injection layers following previous work (Lin et al., 2024). We sample $\mathbf{I}$ with 50 steps of DDIM sampling and set $\eta = 1$ (Song et al., 2021). For structure-rich injection, we set $\tau = 400$ and $C = 600$.

For restart refinement, we set $\sigma_{t_{\min}} = 1.0$, $\sigma_{t_{\max}} = 2.0$, $N = 3$, $S = 5$, where $S$ is the total number of timesteps in the restart backward process. For the restart backward process, we adopt the same noise schedule as the base model, SDXL (Podell et al., 2024), which is:

$$\sigma_{\min} = \sqrt{\frac{\beta_{\min}}{1 - \beta_{\min}}}, \quad \sigma_{\max} = \sqrt{\frac{\beta_{\max}}{1 - \beta_{\max}}}, \tag{14}$$

$$\sigma_t = \sigma_{\min} - (\sigma_{\max} - \sigma_{\min}) \frac{t}{T - 1}, \quad \alpha_t = \frac{1}{1 + \sigma_t^2}, \quad \beta_t = 1 - \alpha_t, \tag{15}$$

where $\beta_{\min} = 0.00085$ and $\beta_{\max} = 0.012$.

For self-recurrence, we set $t'_{\min} = 500$, $t'_{\max} = 900$, $N' = 2$, where $t'_{\max}$ is the self-recurrence starting point, $t'_{\min}$ is the self-recurrence end point, and $N'$ is the number of self-recurrence (Lin et al., 2024). We run most experiments on NVIDIA Tesla V100 GPUs. For FreeControl (Mo et al., 2024), InfEdit (Xu et al., 2024a), and computational efficiency comparisons, we run the experiments on A800 GPUs.

For any input condition image $\mathbf{I}^{\text{struct}}$, we preprocess it with a dilation and unsharp masking operation. Specifically, we binarize the image, perform a distance transform operation to detect the minimum line width $w$. If $w_{\min} \leq w \leq w_{\max}$, we dilate $\mathbf{I}^{\text{struct}}$ with kernel size $k^e$. On the other hand, if the inverted image meets the standard, we erode $\mathbf{I}^{\text{struct}}$. We set $w_{\min} = 25$, $w_{\max} = 50$ and $k^e = 10$. Then we perform unsharp masking $(1 + \gamma) \cdot \mathbf{I}^{\text{dilate}} - \gamma \cdot B$ to modify the dilated (eroded) image, where $\mathbf{I}^{\text{dilate}}$ denotes the dilated (eroded) input condition image, $\gamma = 50$, and $B$ is the Gaussian blur operation with blur radius $r = 3$. We empirically find the two operations beneficial for highlighting object boundaries and improving structure preservation.

## E.2 DATASET DETAILS

We construct our dataset based on the conditional generation datasets from Ctrl-X (Lin et al., 2024) and FreeControl (Mo et al., 2024). Specifically, for conditions canny edge, depth map, normal map, HED edge, and scribble drawing, we select condition-prompt pairs from both datasets and merge them. We collect a total of 15 condition images per condition and form 22 condition-prompt pairs for canny edge and 21 pairs for each of the remaining four conditions.

For human pose and segmentation map, since both original datasets contain limited examples, we supplement them by collecting additional human pose images from the web and segmentation masks from the ADE20K (Zhou et al., 2017) dataset. We obtain 15 images for each of these two conditions and pair them with text prompts using a combination of templates and hand annotation, resulting in 21 image-prompt pairs for human pose and 23 for segmentation mask.

(11/30) Please select the best image, considering its structure alignment with the Input Image, semantic consistency with the Input Text, and visual quality.

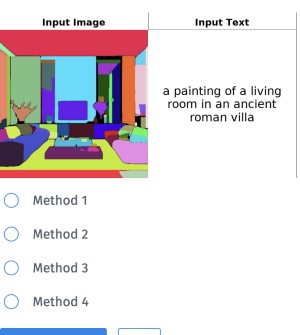

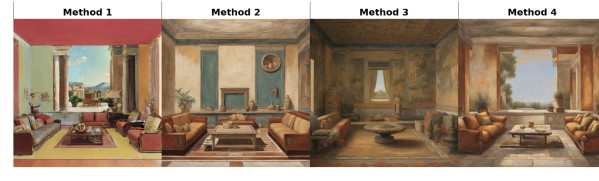

○ Method 1

○ Method 2

○ Method 3

○ Method 4

Next ⌃

Figure 13: **Screenshot of the user study interface.** Participants are presented with the inputs and asked to select the best result from four randomly shuffled candidates.

### E.3 USER STUDY DETAILS

We hereby provide the detailed protocol used for our subjective user study. For each case, participants with relevant expertise are asked to select the best image from four anonymous, randomly shuffled results according to a holistic criterion. The instruction provided in the questionnaire is: *Please select the best image, considering its structural alignment with the Input Image, semantic consistency with the Input Text, and visual quality.* Fig. 13 shows the interface of the user study.

### E.4 COMPUTATIONAL EFFICIENCY EXPERIMENT DETAILS

We evaluate the baselines on our dataset to compare their average inference time and memory usage. Specifically, we implement FreeControl (Mo et al., 2024), Ctrl-X (Lin et al., 2024), and our method using SDXL 1.0 (Podell et al., 2024) checkpoints. For InfEdit (Xu et al., 2024a), we utilize the LCM Dreamshaper v7 (Luo et al., 2023) checkpoint (based on SD1.5), as it is the only model provided in their official codebase. To ensure a fair comparison, we generate $1024 \times 1024$ images using 50 sampling steps for all methods.

## F ADDITIONAL EXPERIMENTAL RESULTS

### F.1 COMPUTATIONAL EFFICIENCY

Complementing the overall efficiency comparison against prior works in Tab. 1, we further analyze the specific runtime contribution of each module within our pipeline in Tab. 3. The SRI module, which dominates the computation (85.1%), represents the core injection framework responsible for handling both condition and appearance image features. Notably, the additional latency introduced by the RR and ARP modules is marginal, accounting for only 6.8% and 8.1% of the total inference time, respectively. Despite their low computational overhead, these components play a critical role in significantly enhancing image quality, as evidenced by the ablation study in Tab. 2.

Table 3: Proportion of inference time consumed by each module of our method.

| Module | Percentage of Inference Time |
| --- | --- |
| SRI | 85.1% |
| RR | 6.8% |
| ARP | 8.1% |

Table 4: **Additional quantitative comparison of controllable T2I.** Our method consistently surpasses all training-free baselines in structure preservation, image-text alignment, and visual diversity. The best results are in **bold**, and the second best are underlined.

| Method | Self-sim ↓ | CLIP ↑ | LPIPS ↑ | Dream-Sim ↓ | Image-Reward ↑ | HPSv2 ↑ |
|---|---|---|---|---|---|---|
| ControlNet (Zhang et al., 2023) | 0.067 | 0.309 | 0.701 | 0.509 | 0.298 | 0.285 |
| T2I-Adapter (Mou et al., 2024) | 0.116 | 0.287 | 0.728 | 0.636 | -0.050 | 0.261 |
| SDEdit (Meng et al., 2022) | 0.154 | 0.259 | 0.315 | 0.734 | -1.374 | 0.189 |
| P2P (Hertz et al., 2023) | 0.197 | 0.251 | 0.266 | 0.724 | -1.786 | 0.168 |
| PnP (Tumanyan et al., 2023) | 0.157 | 0.256 | 0.151 | 0.724 | -1.789 | 0.168 |
| InfEdit (Xu et al., 2024a) | 0.135 | 0.296 | 0.357 | 0.636 | -0.202 | 0.244 |
| FreeControl (Mo et al., 2024) | 0.116 | 0.320 | **0.667** | 0.626 | 0.554 | 0.285 |
| Ctrl-X (Lin et al., 2024) | 0.104 | 0.315 | 0.650 | 0.579 | 0.291 | 0.283 |
| **Ours** | **0.096** | **0.322** | 0.662 | **0.558** | **0.897** | **0.313** |

## F.2 Additional Qualitative Results

We provide additional qualitative comparisons with baselines in Fig. 19 and additional qualitative results for a broader range of condition types in Fig. 20. Our method demonstrates strong generation performance across both common and challenging conditions. It also handles diverse and complex text prompts effectively. As a training-free approach, it generalizes effortlessly to various in-the-wild conditions without any additional training cost, producing high-quality outputs. This level of zero-shot generalization is often unattainable for training-based methods.

## F.3 Additional Quantitative Results

Since T2I-Adapter-SDXL (Mou et al., 2024) supports only four (canny, depth, normal, and pose) out of the seven condition types in our dataset, we further conduct a quantitative comparison limited to these four types. As shown in Tab. 4, our method outperforms all baselines across almost every metric. Notably, these metrics jointly assess both structure preservation (e.g., DINO self-similarity (Tumanyan et al., 2022), DreamSim (Fu et al., 2023)) and generation quality (e.g., ImageReward (Xu et al., 2023a), HPSv2 (Wu et al., 2023)), highlighting the effectiveness of our approach.

## F.4 Additional Ablation Study

We present additional ablation studies on key components of our proposed method to validate our design choices. The results are shown in Fig. 14, Fig. 15, and Fig. 16.

**Structure-Rich Injection.** As a complementary study to the SRI ablation presented in Sec. 5.3, we further investigate the choice of constants in the case of constant injection. Specifically, we evaluate the effects of the injection schedule $g(t) = C$ across various $C$ values. As shown in Fig. 14, lower $C$ values result in severe conditional leakage due to a pronounced domain gap (*e.g.* $C = 0$). In contrast, higher values of $C$ (*e.g.* $C = 800$) produce more natural appearances with higher fidelity but compromise structural control. Empirically, $C = 600$ achieves the best balance between appearance fidelity and structure control, significantly outperforming the synchronous injection baseline by enhancing structural alignment, suppressing condition leakage and reducing visual artifacts simultaneously.

**Appearance-Rich Prompting.** Fig. 15 demonstrates the effectiveness of appearance-rich prompting in enhancing semantic alignment between the structural condition and the appearance image. This strategy helps recover missing semantic elements (*e.g.*, "building" in row 1 and "hand" in row 3), significantly reducing visual artifacts and improving the overall quality of the generated images.

**Restart Refinement.** Fig. 16 illustrates the efficacy of restart refinement in mitigating visual artifacts (*e.g.*, the duplicated eye on the rabbit's body and the incorrect eyes in the husky's background). Additionally, it alleviates condition leakage under abstract conditions (*e.g.*, pose), further improving generation fidelity.

**The number of restart iterations** $N$**.** We also conduct an ablation study of restart iterations $N$. As shown in Fig. 17, setting $N = 1$ is not adequate for suppressing visual artifacts, and both $N = 3$ and $N = 5$ yield high-quality outputs. Consequently, we set $N = 3$ for optimal visual quality and computational efficiency.

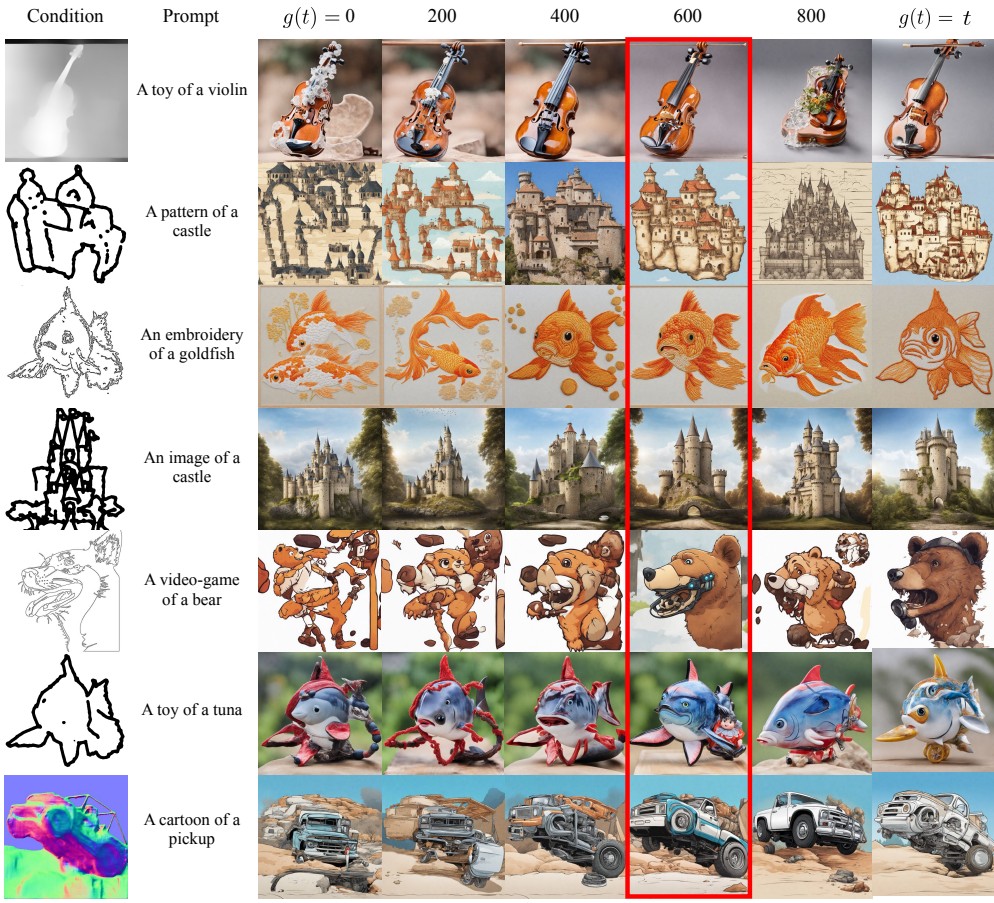

Figure 14: **Additional ablation of Structure-Rich Injection (SRI)**. For asynchronous injection $g(t) = C$, lower $C$ suffers from conditional leakage, while higher values improve appearance fidelity at the cost of structural control. The optimal trade-off is achieved at $C = 600$, outperforming the synchronous schedule ($g(t) = t$).

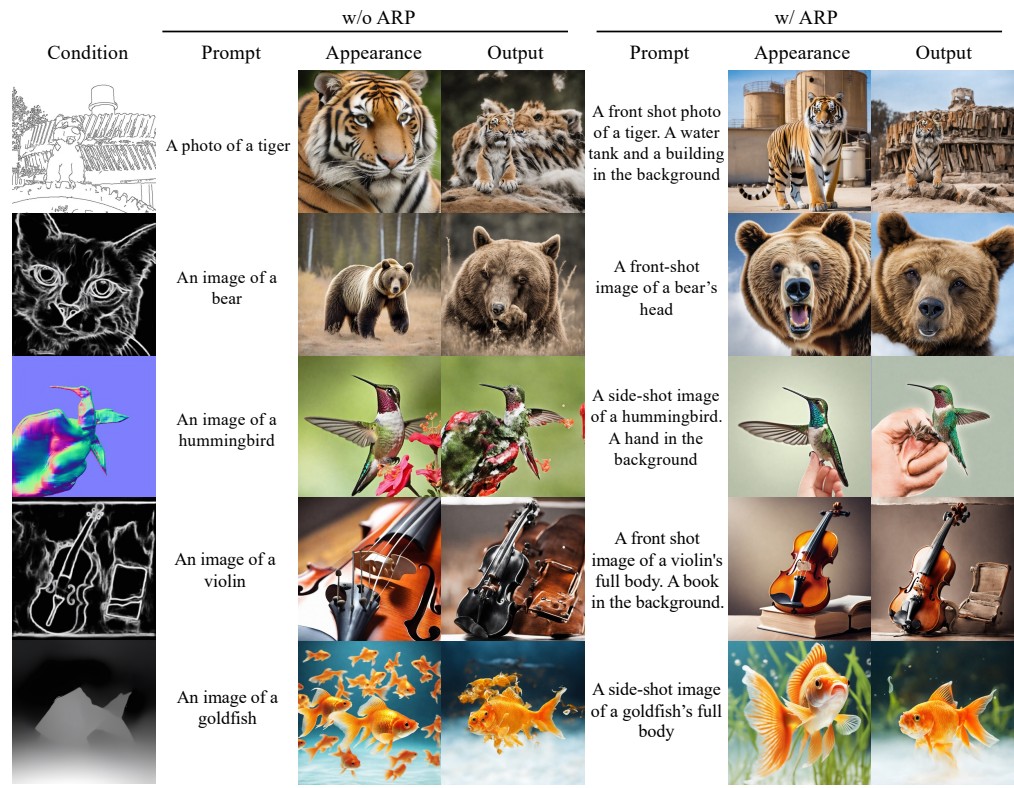

Figure 15: **Additional ablation of Appearance-Rich Prompting (ARP).** This module improves semantic alignment with the condition image by adapting prompts to better capture key visual attributes, thereby mitigating incorrect appearance transfers and reducing artifacts.

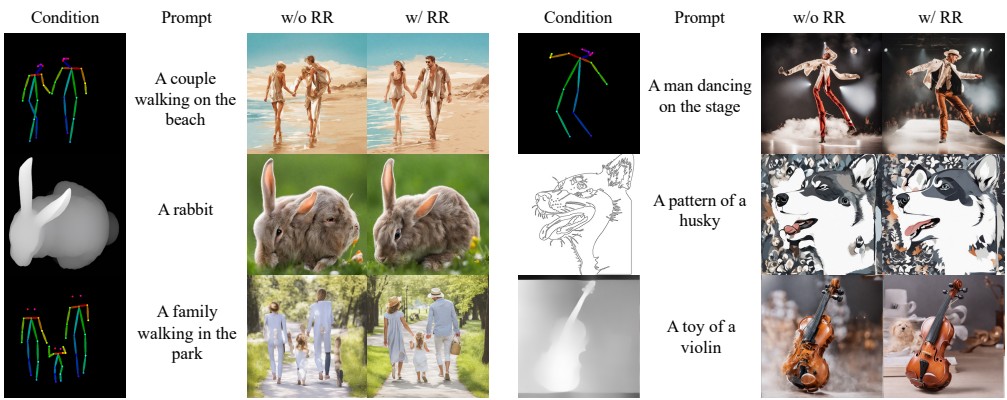

Figure 16: **Additional ablation of Restart Refinement (RR).** This strategy significantly mitigates condition leakage and appearance artifacts, improving generation quality while maintaining structural alignment.

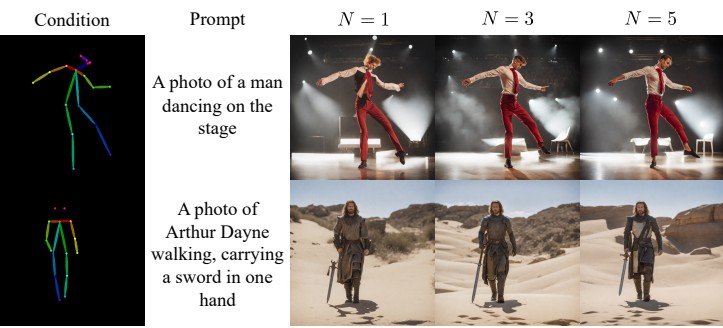

Figure 17: **Additional ablation of restart iterations** $N$. Setting $N = 1$ is not adequate for suppressing visual artifacts, and both $N = 3$ and $N = 5$ yield high-quality outputs. Consequently, we set $N = 3$ for optimal visual quality and computational efficiency.

## F.5 EXPERIMENTS ON DiT-BASED ARCHITECTURES

While our main experiments focused on U-Net-based models (*e.g.*, SDXL, SD1.5) for conditional T2I generation, we conducted exploratory experiments to extend our feature injection paradigm to the Diffusion Transformer (DiT) architecture, specifically FLUX (Labs, 2024). We adopted two distinct strategies to select effective injection layers. First, following the recent analysis by Avrahami et al. (2025), we injected condition features exclusively into the identified *"vital layers"*[1] of FLUX. As shown in Fig. 18, this strategy resulted in negligible structural alignment, indicating that the control signal was insufficient to modulate the deep multimodal attention layers of DiT. To strengthen the control, we subsequently attempted feature injection across all 56 layers. However, this setting resulted in severe condition leakage: the model over-prioritized the condition features, reproducing only the coarse structure of the input while failing to generate coherent textures, leading to significantly degraded appearance quality.

Identifying the optimal injection layers for structure and appearance control within FLUX is a non-trivial task, given the combinatorial search space of $2^{56}$ possible subsets and the complex text–image interactions inherent in its multimodal attention layers. Designing a robust, training-free structure-and-appearance control framework for DiT requires extensive investigation that lies beyond the scope of this work. We therefore leave this exploration to future work.

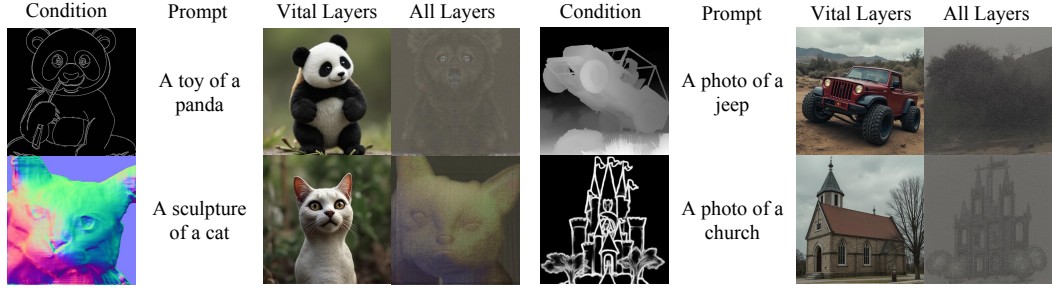

Figure 18: **Results of feature injection for structure control in DiT.** Injecting vital layers (Avrahami et al., 2025) results in inadequate structure alignment, whereas injecting all layers leads to severe condition leakage.

---

[1] The authors of StableFlow (Avrahami et al., 2025) identified nine vital layers for training-free image editing in FLUX through removal-influence analysis. They are layers 0, 1, 17, 18, 25, 28, 53, 54, 56.

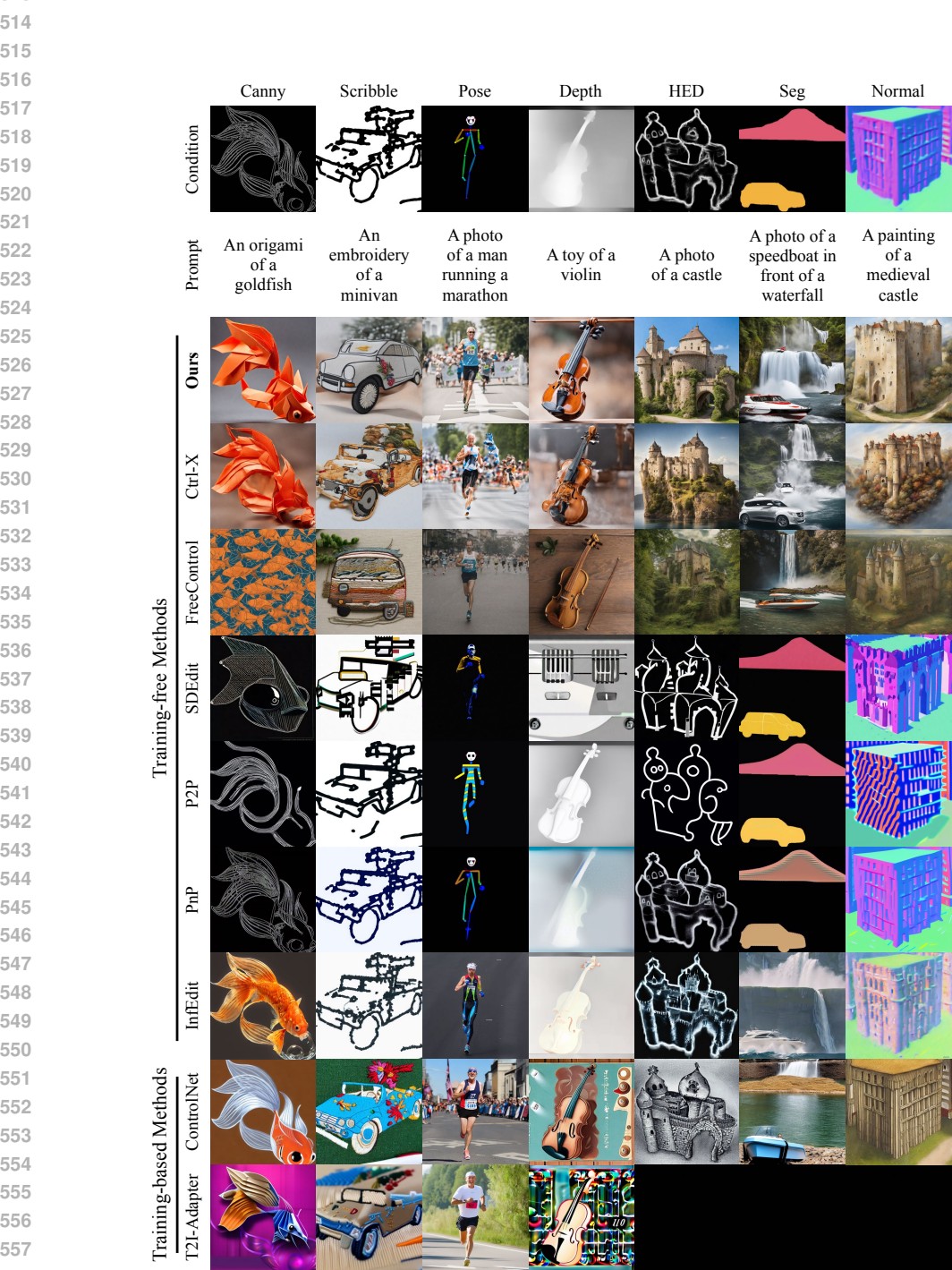

Figure 19: **Qualitative comparison with existing methods.**

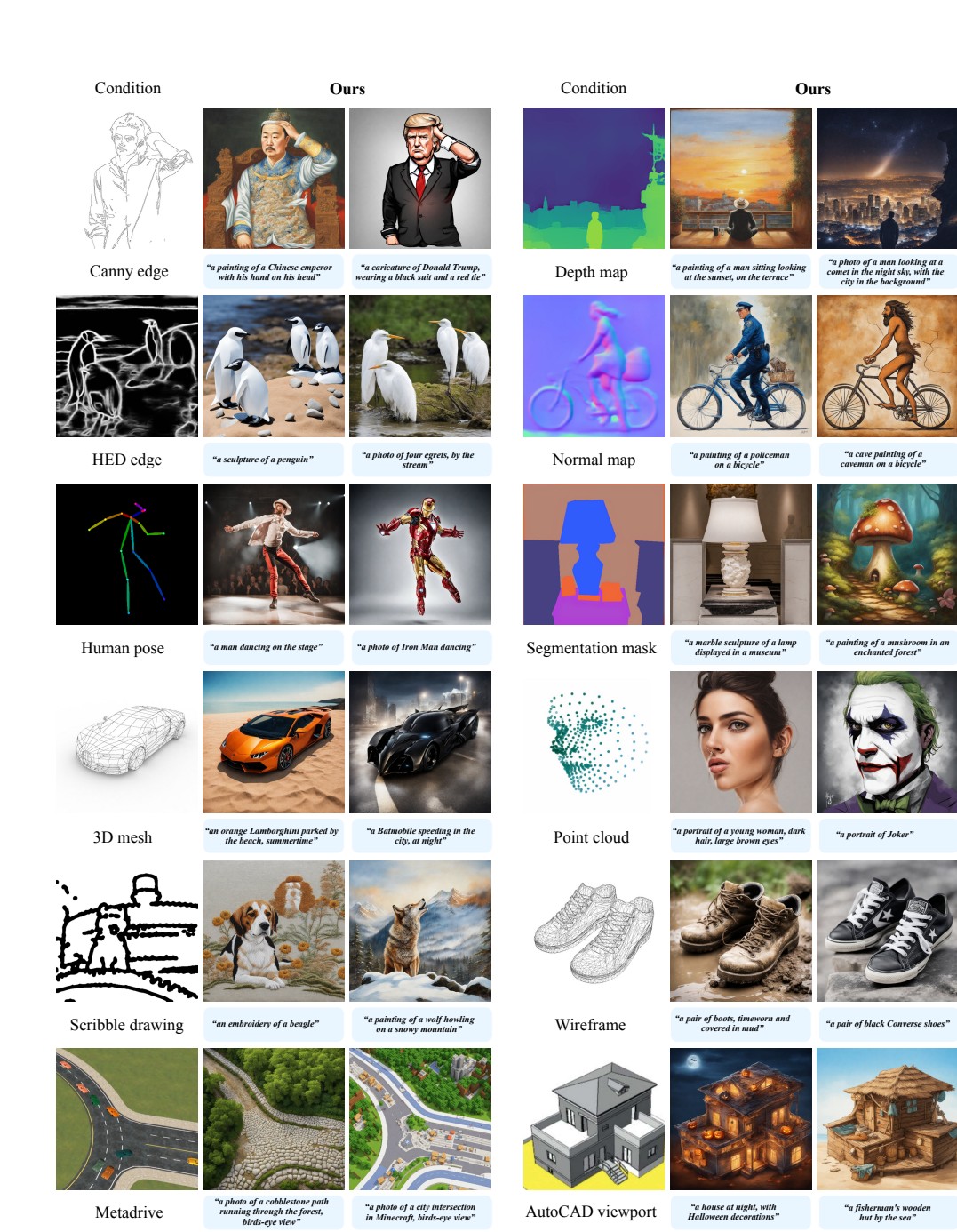

Figure 20: **Qualitative results for more control conditions.**

