# OpenReview forum: "Structure- and Appearance-Rich Training-Free Spatial Control for Text-to-Image Generation"
_ICLR.cc/2026/Conference — Submitted to ICLR 2026_

### Official Review · Reviewer_ogmb · 2025-10-28

**Soundness:** 4
**Presentation:** 4
**Contribution:** 3
**Rating:** 8
**Confidence:** 4

**Summary:**

This paper proposes a training-free framework that enables spatial control for text-to-image diffusion models. The authors first identify the sampling schedule of condition features as a key factor limiting the performance of previous methods, and propose using a single medium timestep to address this issue through systematic analysis. In addition, the authors introduce Restart Refinement and Appearance-Rich Prompting to further enhance the generation quality. Extensive experiments demonstrate the effectiveness of the proposed method.

**Strengths:**

1. The empirical analysis and visualization on the sampling schedule of condition features are insightful and interesting. This analysis provides strong motivation for the proposed method and clearly supports its design choices.
2. The experiments sufficiently support the claimed contributions and demonstrate the effectiveness of the proposed approach.
3. The paper is well-structured and easy to follow.

**Weaknesses:**

1. While Restart Refinement and Appearance-Rich Prompting contribute to improved performance, they also introduce additional computational overhead and increase inference time.
2. This paper mainly focuses on UNet-based T2I models, which are somewhat outdated. It would be interesting to explore whether the discovered sampling schedule of condition features remains effective in DiT-based T2I models.

**Questions:**

How does the inference efficiency of the proposed method compare to other training-free approaches, such as Ctrl-X and FreeControl?

---

> ### Author Response · Authors · 2025-11-22
> **Response to Reviewer ogmb**
>
> We thank the reviewer for the positive evaluation of our work, noting it as insightful, interesting, and effective, and for your thorough and constructive comments. Below, we address the points you raised one by one and have revised our manuscript using red text.
>
> **W1. RR, ARP, additional computational overhead**
>
> **A1.** We have analyzed the inference-time breakdown of our three modules and found that RR and ARP account for only 6.8% and 8.1% of the total runtime, respectively. Thus, they introduce only marginal computational overhead while significantly improving image quality, as validated in our ablation study (Tab. 2 in the main revised manuscript). We have updated Appendix F.1 to include this analysis.
>
> | Module | Percentage of Inference Time |
> | ------ | ---------------------------- |
> | SRI    | 85.1%                        |
> | RR     | 6.8%                         |
> | ARP    | 8.1%                         |
>
> **W2. DiT-based T2I models**
>
> **A2.** Thank you for your suggestion. Training-free controllable diffusion models on DiT architectures are still in their early stages. Existing work [a,b,c,d,e] mainly focuses on editing, which is simpler than fine-grained structure control, as editing preserves most regions of the input image, whereas structure control requires precise manipulation of spatial structure. As revealed in prior work [a], DiT models introduce additional challenges: (i) they are significantly deeper (e.g., 56 layers in FLUX); (ii) their features are less interpretable than U-Net features. As a result, existing DiT editing work [a,b,c] uses removal influence to empirically investigate the role of each layer in DiT.
>
> Following the reviewer's suggestion and the strategy in [a], we conducted additional experiments on FLUX by injecting condition features into the empirically identified "vital layers.". As shown in Fig. 18 (Sec. F.5) of the appendix in the revision, we found that the resulting control strength was insufficient, as the generated images failed to follow the structural condition. Therefore, to improve control strength, we attempted to inject at all layers, which, however, led to severe condition leakage. We hypothesize that this failure mainly arises from the deeper layers of DiT, and that identifying the injection layers that effectively control structural changes is a non-trivial task. We hope future work can further explore the capabilities of DiT in this direction.
>
> We have included detailed qualitative results and discussion in Sec. F.5 of the supplementary manuscript.
>
>
> **Q1. the inference efficiency of the proposed method**
>
> **AQ1.** Thanks for your suggestion. We have included the inference efficiency of our method and baselines in Sec. 5.2 of the revised manuscript. Our method achieves the fastest inference speed (18.79s per image) among strong baselines including Ctrl-X and FreeControl, confirming its computation efficiency.
>
> | Method     | Time (s) | Memory (GB) |
> |------|-----|-----|
> | InfEdit  | 31.84  | 52.44  |
> | FreeControl  | 781.57 | 55.46  |
> | Ctrl-X  | 19.37 | 18.79  |
> | **Ours**  | **18.79**  | **18.77**  |
>
> [a] *Stable Flow: Vital Layers for Training-Free Image Editing*, CVPR 2025
>
> [b] *QK-Edit: Revisiting Attention-based Injection in MM-DiT for Image and Video Editing*, ICCV 2025
>
> [c] *FreeFlux: Understanding and Exploiting Layer-Specific Roles in RoPE-Based MMDiT for Versatile Image Editing*, ICCV 2025
>
> [d] *FluxSpace: Disentangled Semantic Editing in Rectified Flow Models*, CVPR 2025
>
> [e] *DiT4Edit: Diffusion Transformer for Image Editing*, AAAI 2025

---

### Official Review · Reviewer_ZU36 · 2025-10-30

**Soundness:** 2
**Presentation:** 3
**Contribution:** 2
**Rating:** 4
**Confidence:** 2

**Summary:**

This paper proposes a training-free framework for enhancing spatial control in text-to-image diffusion models.

The authors diagnose key issues in prior feature-injection-based approaches (e.g., Ctrl-X), including structure misalignment, condition leakage, and visual artifacts, attributing them to the inadaptiveness of the sampling schedule.

To address these problems, the paper introduces three modules: Structure-Rich Injection (SRI), Restart Refinement (RR), and Appearance-Rich Prompting (ARP).

The proposed framework achieves state-of-the-art results across various conditional modalities (canny, depth, pose, segmentation, etc.) without any training, outperforming both training-based (e.g., ControlNet, T2I-Adapter) and training-free (e.g., FreeControl, Ctrl-X) methods in structure fidelity and visual realism.

**Strengths:**

The paper provides a convincing diagnosis of why training-free spatial control methods often fail, linking it to the inadaptiveness of the injection schedule.

The Structure-Rich Injection mechanism is simple, training-free, and computationally efficient thanks to feature caching.

The proposed SRI, RR, and ARP components are complementary and can be easily integrated with existing diffusion models such as SDXL.

The authors evaluate across diverse condition modalities, using both objective metrics (DreamSim, ImageReward, HPSv2) and qualitative analysis, consistently demonstrating superior results.

Despite being training-free, the method rivals or even surpasses fine-tuned approaches like ControlNet, highlighting impressive generality and real-world potential.

**Weaknesses:**

The analysis of the sampling schedule remains empirical; the paper lacks a formal justification or theoretical insight into why mid-timestep injection yields optimal results.

Although the paper discusses the trade-off between structural fidelity and visual appearance, it does not deeply explore the semantic interactions in the feature space.

The claim of ‘applicability to arbitrary pretrained diffusion models’ is mainly demonstrated on SDXL, with limited evidence for other architectures (e.g., DiT, LDM).

Restart Refinement improves appearance quality but occasionally compromises structure alignment; optimal iteration count and noise levels are empirically chosen.

The Appearance-Rich Prompting component depends on the quality and consistency of the employed multimodal LLM, which may limit reproducibility.

Absence of human evaluation: The evaluation relies solely on reward-based metrics that approximate human preference; no actual human study is presented.

**Questions:**

Could the authors provide a more theoretical explanation or empirical ablation supporting why medium timesteps (around 600) yield optimal structure-appearance balance?

---

> ### Author Response · Authors · 2025-11-22
> **Response to Reviewer ZU36 (1/3)**
>
> We thank the reviewer for recognizing our convincing diagnosis of prior methods’ limitations, as well as the simplicity, efficiency, and superior performance of our method, and for your thorough and constructive comments. Below, we address the points you raised one by one and have revised our manuscript using red text.

---

> ### Author Response · Authors · 2025-11-22
> **Response to Reviewer ZU36 (2/3)**
>
> **W1 and Q1. theoretical insight or empirical ablation**
>
> **A1.** We thank the reviewer for the suggestion. In Fig. 14, we've provided an empirical ablation of constant injection timesteps, showing that step 600 achieves a better trade-off. We attempted to provide theoretical insight into why mid-timestep injection works best; however, since diffusion features evolve in a high-dimensional latent space, providing a formal justification is extremely challenging and non-trivial. Following many prior works (PnP, FreeControl, Ctrl-X), we instead rely on empirical feature-space examinations, such as PCA visualizations, to explore patterns in the diffusion feature space. In this spirit, Sec. 3 presents a multi-perspective analysis that consistently highlights the mid-timestep region as yielding a better balance between structure and appearance. We appreciate the reviewer for highlighting this point and have discussed it in Sec. 6 as a direction for future investigation.
>
> **W2. semantic interactions in feature space**
>
> **A2.** We appreciate the reviewer's comment on semantic interactions in the feature space, which is also a motivation and a focus point of our approach. In Sec. 3, we focus on the tradeoff between structural fidelity and visual appearance, revealing the evolving interaction with both quantitative and qualitative studies. As shown in Fig. 2, throughout the denoising process, the enhanced structure alignment comes at the cost of a wider domain gap. The PCA visualization in Fig. 3 further reveals the mechanism of the interaction: As the modality-specific details emerge in the late stage (small $t$), the original semantic clues become more vague, and the visual quality of the image is downgraded. This trend is further confirmed in the ablation study of SRI schedules (Fig. 7), as schedules ending at later timesteps get worse appearance-related scores (HPSv2 and ImageReward). We hope we have correctly understood the reviewer’s point and welcome any further clarification.
>
> **W3. other architectures**
>
> **A3.**
> We appreciate the reviewer for raising this problem. We would like to clarify that we did not state that our method can be applied to “arbitrary pretrained diffusion models”. If we understand correctly, the mentioned “LDM” refers to Latent Diffusion Models, which typically adopt UNet architectures (e.g., SD1.5, SDXL). These models benefit from well-studied, interpretable diffusion features that enable training-free structure control, as demonstrated in prior works (FreeControl, Ctrl-X). To prevent ambiguity, we have revised Sec. 5.4 to explicitly specify the scope as "U-Net-based architectures" in our main experiments.
>
> We acknowledge the growing importance of the DiT framework. However, training-free structural control on DiT remains an open challenge. Existing work [a,b,c,d,e] mainly focuses on editing, which is simpler than fine-grained structure control, as editing typically involves semantic-level replacements, whereas structure control requires more precise manipulation of spatial structure. As revealed in prior work [a], DiT models introduce additional challenges: (i) they are significantly deeper (e.g., 56 layers in FLUX); (ii) their features are less interpretable than UNet features. As a result, existing DiT editing work [a,b,c] uses removal influence to empirically investigate the role of each layer in DiT.
>
> Following the reviewer's suggestion and the strategy in [a], we conducted additional experiments on FLUX by injecting condition features into the empirically identified "vital layers.". As shown in Fig. 18 (Sec. F.5) of the appendix in the revision, we found that the resulting control strength was insufficient, as the generated images failed to follow the structural condition. Therefore, to improve control strength, we attempted to inject at all layers, which, however, led to severe condition leakage. We hypothesize that this failure mainly arises from the deeper layers of DiT, and that identifying the injection layers that effectively control structural changes is a non-trivial task. We hope future work can further explore the capabilities of DiT in this direction.

---

> ### Author Response · Authors · 2025-11-22
> **Response to Reviewer ZU36 (3/3)**
>
> **W4. effect of RR on structural alignment; choice of iteration count and noise level**
>
> **A4.**
> Although the RR module introduces a minor compromise on structural alignment, our quantitative ablation (Tab. 2) shows that this impact is very limited. However, the resulting substantial improvement in appearance quality is critical for the final output (Fig. 8, Fig. 16), justifying the effectiveness of this module in achieving the structure-appearance trade-off.
>
> Regarding the hyperparameters of RR:
> - Iteration count: We conducted an additional ablation study to validate our choice ($N=3$). As shown in Fig. 17 (Sec. F.4) of the Appendix in the revision, our selected iteration count offers optimal performance and efficiency.
> - Noise level: We adopted the settings theoretically established in prior work [f] , which proved robust and effective in our experiments.
>
> We have revised Section F.4 of the Appendix to include these additional experiments and discussions. Please refer to it for a detailed analysis.
>
> **W5. ARP reproducibility**
>
> **A5.** When designing ARP, we took into account that multimodal LLMs may occasionally produce errors, so we implemented safeguard mechanisms to ensure that the ARP module is never worse than the user input plain prompt and is usually strictly better, such as "If the input dictionary is empty or invalid, return an empty dictionary {}.", "If the input dictionary is empty or invalid, return the input sentence unchanged." (see Supplementary Sec. D.2 for more details). The `ARP prompt.txt` file in our supplementary material contains the specific prompts used.
>
> Additionally, to ensure reproducibility, all quantitative comparisons and ablation studies were repeated three times, and we report the mean results across these runs. Therefore, we believe that our results are statistically reliable.
>
> **W6. Absence of human evaluation**
>
> **A6.** Thank you for your suggestion. During the rebuttal period, we conducted a user study to assess human preference; the results and details have been included in Sec. 5.2 (Tab. 1) and Supplementary Sec. E.3. We randomly sampled 30 cases from the dataset and compared our method against the three strongest training-free baselines: Ctrl-X, FreeControl, and InfEdit. Participants were asked to choose the image that ranks best in structural alignment with the condition image, semantic consistency with the prompt, and visual quality. We collected responses from 40 participants, all with backgrounds in computer vision, ensuring that their judgments were informed and reliable. 56.25% of human users prefer our method over the baselines, further validating the effectiveness of our approach.
>
> | Method      | Preference Rate |
> | ----------- | --------------- |
> | InfEdit     | 11.42%          |
> | FreeControl | 10.67%          |
> | Ctrl-X      | 21.67%          |
> | **Ours**        | **56.25%**          |
>
>
> [a] *Stable Flow: Vital Layers for Training-Free Image Editing*, CVPR 2025
>
> [b] *QK-Edit: Revisiting Attention-based Injection in MM-DiT for Image and Video Editing*, ICCV 2025
>
> [c] *FreeFlux: Understanding and Exploiting Layer-Specific Roles in RoPE-Based MMDiT for Versatile Image Editing*, ICCV 2025
>
> [d] *FluxSpace: Disentangled Semantic Editing in Rectified Flow Models*, CVPR 2025
>
> [e] *DiT4Edit: Diffusion Transformer for Image Editing*, AAAI 2025
>
> [f] *Restart Sampling for Improving Generative Processes*, NeurIPS 2023

---

### Official Review · Reviewer_MMMv · 2025-10-31

**Soundness:** 3
**Presentation:** 2
**Contribution:** 2
**Rating:** 4
**Confidence:** 3

**Summary:**

The paper presents a novel, training-free framework for spatial control in text-to-image (T2I) generation. The authors propose a method that enhances control over structure and appearance during the generation process, addressing limitations in existing methods. By analyzing the temporal dynamics of diffusion features, they discover that the sampling schedule for condition features plays a crucial role in balancing structure alignment and visual quality. Their solution decouples the condition feature sampling from the denoising process, resulting in improved structural preservation and visual fidelity. The framework consists of three key components: Structure-Rich Injection (SRI)**, **Restart Refinement (RR), and Appearance-Rich Prompting (ARP). Extensive experiments demonstrate that this approach outperforms state-of-the-art (SOTA) methods across a variety of zero-shot conditioning scenarios.

**Strengths:**

1. This approach eliminates the need for additional fine-tuning, making it highly adaptable and efficient for various models and conditions.

2. By identifying and addressing the limitations of existing feature injection schedules, the authors provide a principled way to sample condition features, significantly improving structure preservation and visual fidelity.

**Weaknesses:**

1. The success of the method relies heavily on empirical observations regarding the optimal timesteps for feature injection. While the paper provides a comprehensive analysis, the method's effectiveness could vary depending on the specific nature of the condition images used, requiring further validation across diverse datasets and domains.

2. The framework introduces additional steps such as the Restart Refinement (RR) schedule and Appearance-Rich Prompting (ARP) strategy. While this enhances performance, it could lead to increased computational requirements.

**Questions:**

The paper uses multiple evaluation metrics, but have the authors considered designing a comprehensive metric that would evaluate the balance between structure alignment and visual quality?

---

> ### Author Response · Authors · 2025-11-22
> **Response to Reviewer MMMv**
>
> We thank the reviewer for recognizing the novelty, adaptability, efficiency, and significant improvements of our work and for your thorough and constructive comments. Below, we address the points you raised one by one and have revised our manuscript using red text.
>
> **W1. diverse datasets and domains.**
>
> **A1.** Thanks for your suggestion of conducting further validation across diverse datasets and domains. In fact, we have already collected a **large, balanced** dataset comprising **equal-sized subsets** across **seven diverse** conditioning types, as detailed in Sec. 5.1 and Sec. E.2. These conditions include canny edge, depth map, HED edge, normal map, human pose, segmentation mask, scribble drawing, making them strong representatives of diverse scenarios. We believe that the results on this dataset are statistically meaningful, as our method consistently achieves the best performance across all categories.
>
> In addition, we've also presented results and comparisons on 12 more condition types in Fig. 19 and Fig. 20, including canny edge, depth map, HED edge, normal map, human pose, segmentation mask, 3D mesh, point cloud, scribble drawing, wireframe, metadrive, AutoCAD viewport, which further demonstrates the robustness of our injection schedule across domains. We believe these extensive experiments are sufficient to validate the generality of our approach, and we plan to explore even more diverse domains in future work.
>
> **W2. RR and ARP could lead to increased computation**
>
> **A2.** We have analyzed the inference-time breakdown of our three modules and found that RR and ARP account for only 6.8% and 8.1% of the total runtime, respectively. Thus, they introduce only marginal computational overhead while significantly improving image quality, as validated in our ablation study (Tab. 2 in the main manuscript). We have updated Appendix F.1 to include this analysis.
>
> | Module | Percentage of Inference Time |
> | ------ | ---------------------------- |
> | SRI    | 85.1%                        |
> | RR     | 6.8%                         |
> | ARP    | 8.1%                         |
>
>
> **Q1. a comprehensive metric**
>
> **AQ1.** Thank you for your suggestion.
> Evaluating generated images involves multiple aspects, such as structural alignment and visual quality, and current metrics are typically designed to assess these separately. We find this disentangled evaluation framework especially helpful for our task, as it allows us to better understand the effects of controlling both structure and appearance. While it could be possible to combine metrics into a single comprehensive score, for example, via a weighted sum, this could introduce new challenges in interpreting trade-offs. We agree that developing a unified metric is an interesting direction for future work.

---

### Official Review · Reviewer_jVYY · 2025-11-01

**Soundness:** 2
**Presentation:** 2
**Contribution:** 2
**Rating:** 4
**Confidence:** 3

**Summary:**

The paper proposes a training-free framework for spatially controlled text-to-image generation.
It decouples the sampling schedule of condition features from the denoising process. The method achieves improved structural alignment, reduced condition leakage, and higher visual fidelity across diverse control modalities without retraining.

**Strengths:**

-- The paper identifies a limitation in prior work: fixed timestep injection fails to balance structural fidelity and domain alignment, validated via KL divergence and L2 distance curves in Fig. 2.

-- Restart Refinement (RR) reduces condition leakage and visual artifacts while preserving structure, as qualitatively shown in Fig. 8 and quantitatively in Table 1. ARP also demonstrate improvements.

--The method outperforms some training-free and training-based baselines on reward-model metrics aligned with human preferences.

**Weaknesses:**

-- The noise sampling process “from the perturbation kernel” lacks specification of distribution or variance schedule.

-- The number of restart iterations N is mentioned but its selection criterion or default value is omitted.

-- Hyperparameters for baselines (e.g., inversion steps for DDIM in Ctrl-X) are not detailed, though results depend on them. Does the method keep the same hyperparameters to the baseline?

-- No timing or memory measurements are provided for the proposed method versus baselines, despite caching claims.

-- Figure 4 is not informative and not easy to understand.

**Questions:**

Please see the weakness.

---

> ### Author Response · Authors · 2025-11-22
> **Responses to Reviewer jVYY**
>
> We thank the reviewer for recognizing our contributions in addressing the limitations in prior work and our improvement, and for your thorough and constructive comments. Below, we address the points you raised one by one and have revised our manuscript using red text.
>
> **W1. specification of distribution or variance schedule**
>
> **A1.** The noise sampling schedule of the restart forward process follows the original noise schedule of the base model, SDXL [a]:
>
> $$
> \sigma_{\min} = \sqrt{\frac{\beta_{\min}}{1-\beta_{\min}}}, \quad \sigma_{\max} = \sqrt{\frac{\beta_{\max}}{1-\beta_{\max}}},
> $$
>
> $$
> \sigma_t = \sigma_{\min} - \left(\sigma_{\max}-\sigma_{\min}\right)\frac{t}{T-1}, \quad\alpha_t = \frac{1}{1+\sigma_t^2}, \quad\beta_t = 1 - \alpha_t,
> $$
>
> where $\beta_{\min} = 0.00085$ and $\beta_{\max} = 0.012$. In the restart forward process, we set $\epsilon_{t_{min}} = 1.0$, $\epsilon_{t_{max}} = 2.0$. We have included all needed implementation details in Appendix E.1.
>
> **W2. iterations N ... selection criterion or default value**
>
> **A2.** We would like to kindly note that this was already included in Appendix E.1 of our original submission, where we set $N=3$. We have also conducted an additional ablation to validate the choice and included the results in Appendix F.4 of the revised paper. As shown in Fig. 17, setting $N=1$ is not adequate for suppressing visual artifacts, and both $N=3$ and $N=5$ yield high-quality outputs. Consequently, we set $N=3$ for optimal visual quality and computational efficiency.
>
> **W3. same hyperparameters to the baseline**
>
> **A3.** Yes. We keep the same hyperparameters for the baselines to ensure a fair comparison. We use 50 denoising steps and 50 inversion steps for all baselines. For all other hyperparameters, we follow the settings of their respective official implementations. The implementation details for all baselines have been included in Sec. 5.1 of the revised paper.
>
> **W4. timing and memory measurements for the proposed method versus baselines**
>
> **A4.** Thanks for your suggestion. We have included the inference time of our method and baselines in Sec. 5.2 of the revised manuscript. Our method achieves the fastest inference speed (18.79s per image) and lowest memory cost (18.77GB) among strong baselines, confirming its computation efficiency.
>
> | Method     | Time (s) | Memory (GB) |
> |------|-----|-----|
> | InfEdit  | 31.84  | 52.44  |
> | FreeControl  | 781.57 | 55.46  |
> | Ctrl-X  | 19.37 | 18.79  |
> | **Ours**  | **18.79**  | **18.77**  |
>
> **W5. Clarification on Figure 4**
>
> **A5.**
> We have revised Sec. 4 and the caption of Fig. 4 to better align the textual description with the visual elements and improve the logical flow. We have highlighted these changes in red in the revised PDF.
> Below is a brief walkthrough of the improved logic flow corresponding to Fig. 4:
> Given a condition image $\mathbf{I}^\text{struct}$ and a prompt $\mathcal{P}$, our method generates an output image $\mathbf{I}$, aligning semantically with $\mathcal{P}$ while preserving the structure of $\mathbf{I}^\text{struct}$. Our framework consists of three key components.
> (i) The **Structure-Rich Injection (SRI) module (blue)** injects structure-rich condition features $\mathbf{f}\_{l,g(t)}^\text{struct}$ and attentions $\mathbf{A}\_{l,g(t)}^\text{struct}$ into the output feature space to enable spatial control (Sec. 4.1).
> (ii) The **Restart Refinement (RR) module (pink)** iteratively adds noise to and denoises $\mathbf{I}$ to refine visual details such as the eyes of the bear (Sec. 4.2).
> (iii) The **Appearance-Rich Prompting (ARP) module (green)** derives an enriched prompt $\mathcal{P}^\text{app}$ based on the semantics of the condition image $\mathbf{I}^\text{struct}$ to generate a reference image $\mathbf{I}^\text{app}$ for appearance transfer (Sec. 4.3).
> Please refer to the revised PDF for the updated caption and detailed exposition.
>
> [a] *SDXL: Improving latent diffusion models for high-resolution image synthesis*, ICLR 2024

---

### Author Response · Authors · 2025-11-22
**General Responses**

We sincerely thank all reviewers for their time and insightful feedback, and for recognizing the value of our work.

Firstly, reviewers commend our *“convincing diagnosis”* of the limitations of prior feature injection schedules (**Reviewer ZU36, jVYY**, **MMMv**). **Reviewer ogmb** emphasizes that our *“insightful and interesting”* analysis strongly motivates the proposed design choices. Building on this foundation, reviewers recognize the *“novelty”* of our method (**Reviewer MMMv**), highlighting its *“simplicity”*, *“efficiency”*, and *“adaptability”* (**Reviewer ZU36, MMMv**). Finally, we are encouraged that all reviewers validate our method's *“superior performance”* over training-free baselines (i.e. improved structural alignment, reduced condition leakage, and greater visual fidelity), as demonstrated by our extensive experiments.

We have revised the manuscript (main text and supplementary) according to the comments, with changes highlighted in **red**. Below, we provide detailed responses to the issues raised by each reviewer.

---

### Author Response · Authors · 2025-12-01
**Response to Reviews and Revision Summary for the Area Chair**

Dear Area Chair,

Thank you for taking the time to evaluate our submission. To facilitate your evaluation, we provide a concise paper summary, a brief overview of reviewer feedback, our rebuttal, and the revisions we have made during the rebuttal period.

**Paper Summary:**

We propose a training-free, structure- and appearance-rich control method for text-to-image generation models. By identifying the limitations of existing feature-injection schedules, we introduce an approach that decouples condition feature sampling from the denoising process, leading to improved structure preservation and visual fidelity. Our method not only outperforms prior training-free baselines but also surpasses fine-tuned approaches, making it highly adaptable and efficient across different models and conditions.

**Review Summary:**

We thank all reviewers for recognizing the novelty and contributions of our work and for providing constructive feedback that helped improve the manuscript. Specifically:

- Novelty & Motivation: All reviewers highlighted our "convincing" diagnosis of the limitations of prior feature injection schedules, and **Reviewer ogmb** praised our analysis as "insightful and interesting", providing strong motivation for our design choices.

- Performance: All reviewers recognized our strong performance over existing baselines. Reviewers noted that our method outperforms baselines (**jVYY**), "significantly improving structure preservation and visual fidelity" (**MMMv**), "even surpasses fine-tuned approaches" (**ZU36**), and "sufficiently support the claimed contributions and demonstrate the effectiveness of the proposed approach" (**ogmb**).

- Potential Future Directions:

1. **Reviewer ogmb** and **ZU36** suggested "it would be interesting to explore" the application of our proposed method on DiT models. In response, we have conducted experiments and provided analysis in Appendix F.5 and Fig. 18 of the revised manuscript. The results indicate that this is a promising direction for future work that could benefit from broader community efforts, and does not affect our claimed contributions.

2. **Reviewer ZU36** suggested providing "a more theoretical explanation or empirical ablation" on the timestep choice. We have included a detailed empirical ablation in Appendix F.4 and Fig. 14, which directly addresses this suggestion. Due to the high dimensionality of the diffusion feature space, formal theoretical justification is extremely challenging and may require broader community efforts. Nevertheless, Sec. 3 analyzes feature patterns, and Sec. 6 highlights this as a promising direction for future work.

**Revision Summary:**

In response to the reviewers’ comments, we carefully revised the paper with changes highlighted in red. We hope these updates adequately address all concerns and improve the manuscript’s clarity and completeness.

1. We revised Sec. 4 and the captions of Fig. 4 to improve clarity (**jVYY, W5**).
2. We revised Sec. 5.1 and Appendix E.1 to provide additional implementation details (**jVYY, W1, W3**; **ZU36, W5**).
3. We revised Sec. 5.2 (added Tab. 1) and Appendices E.3, E.4, and F.1 (added Tab. 3 and Fig. 13) to include computational efficiency analysis and a user study (**jVYY, W4**; **MMMv, W2**; **ZU36, W6**; **ogmb, W1, Q1**).
4. We revised Sec. 5.4 and Appendix F.5 (added Fig. 18) to clarify model adaptability and provide additional experiments on DiT-based architectures (**ZU36, W3**; **ogmb, W2**).
5. We revised Sec. 6 to further discuss limitations and future directions (**ZU36, W1, Q1**).
6. We revised Appendix F.4 (added Fig. 17) to include an ablation on restart iterations (**jVYY, W2**; **ZU36, W4**).

We sincerely appreciate the Area Chair and the reviewers for their thoughtful feedback and the opportunity to improve our work. We hope that the revisions have addressed the concerns raised and have strengthened the quality of the paper.


Sincerely,

Authors of Submission 4127

---

### Meta-Review · Area_Chair_EjEm · 2025-12-30

**Summary:**

This paper explores a training-free text-to-image diffusion model that incorporates varied structural and spatial image conditioning during the generative process. The key insight of the paper concerns the impact of generated image quality as a function of the schedule used to inject features from the conditioning image. Empirically, the paper argues that using a constant injection schedule at an intermediate diffusion timestep yields the best results. The paper also proposes enhanced image prompting and a diffusion restart process to further improve generation quality. Experiments are conducted on several public datasets and demonstrate state-of-the-art performance compared to closely related methods, including Ctrl-X.

**AC Comments:**

The paper received mixed reviews, with one accept (ogmb) and three borderline rejects (jVYY, ZU36, MMMv). While Reviewer ogmb and the other reviewers raise largely overlapping issues, their interpretations of the contributions differ. The key outstanding concerns include:

i) the lack of technical or theoretical insight into the impact of the conditioning injection schedule (MMMv, ZU36);

ii) the absence of analysis relating the choice of schedule to the modality of the conditioning input (MMMv); and

iii) the reliance on additional modules (RR and ARP) to improve performance, which introduce further dependencies and engineering choices (ogmb, ZU36, MMMv).

Additional concerns were raised regarding the generality of the approach to other architectures, including DiTs (ogmb, ZU36).

Beyond these points, AC believes the paper should demonstrate the performance of the method when using only the proposed scheduling strategy, without relying on the ARP and RR modules, which are established techniques and appear tangential to the paper’s main contribution. While some ablation results are presented in Table 2, they do not include comparisons to baselines or cover the full range of evaluation metrics, limiting their ability to highlight the true benefit of the proposed core idea.

Considering the full set of arguments and the likelihood of reviewers changing their scores based on the responses outlined below, AC believes the paper would continue to receive mixed evaluations. In particular, AC considers the concerns raised by reviewers MMMv and ZU36 to be critical and necessary to address for acceptance. Accordingly, AC recommends rejection. The authors are encouraged to revise the paper to address these concerns in a future submission.

**Reviewer Concerns:**

*Reviewer jVYY* seeks clarification on several details omitted from the main paper, including the specification of the noise sampling process in the perturbation kernel, the absence of a study on restart iterations, missing hyperparameter details for the baselines, and the omission of memory and timing measurements.

*Reviewer MMMv* raises two important concerns regarding the study: i) the reliance of the approach on empirical findings, which may be questionable since different types of conditioning inputs could benefit from different injection schedules; and ii) the dependence on additional modules such as restart refinement and appearance-rich prompting, which deviate from the main line of study but appear to be important for achieving strong performance.

*Reviewer ZU36* argues that the paper’s observations regarding the importance of the conditioning injection schedule are purely empirical and lack theoretical grounding, including an analysis of the trade-off between structural fidelity and appearance, as well as the role of restart refinement. The reviewer also questions the generality of the approach to other diffusion architectures, noting that this is not substantiated by the experiments, raises concerns about the strength of the multimodal LLMs used in the ARP module, and points out the absence of human evaluation.

*Reviewer ogmb* raises two concerns that overlap with those of other reviewers, specifically regarding the computational overhead introduced by the RR and ARP modules and the generality of the method to non-UNet-based models, including DiTs.

**Reviewer Scores:**

**Reviewer jVYY:** Authors provide an adequate response with details of the noise sampling process (described in the Appendix of the revised paper), along with ablation studies on the number of restart iterations and other technical details, which are mostly included in the Appendix.

[*AC's thoughts on the response*]
The questions raised by the reviewer appear to be sufficiently addressed by the additional details provided by the authors, and AC therefore believes the reviewer may have raised the score to at least borderline accept.

**Reviewer MMMv:** Authors point to several qualitative results provided in the main paper and the Appendix that demonstrate high-quality generation for varied types of conditioning inputs, including Canny edges, depth maps, and point clouds. Further, to address the additional computational load introduced by the RR and ARP modules, the authors present detailed measurements showing that each module contributes less than 10% of the total inference time.

[*AC's thoughts on the response*]
The reviewer’s question regarding why the same injection schedule may work across all types of conditioning inputs remains important and lacks a sufficiently detailed analysis in the paper. While the qualitative results presented by the authors are interesting, they only partially address this concern. Given that the main insight of the paper centers on identifying an effective injection schedule, AC believes the response does not adequately resolve this issue, and the reviewer is likely to have maintained a score of borderline reject.

**Reviewer ZU36:** To address concerns about the lack of theoretical insight, the authors continue to rely on empirical findings, specifically referencing Fig. 14 and the discussion in Sec. 3. Regarding the generality of the approach, the authors clarify that the method is applicable only to UNet-style diffusion architectures. They also present new experiments using DiT-type models (e.g., FLUX), showing that the method is less effective even when conditioning features are injected at all network layers. In addition, the authors present a new human study on 30 generated examples, reporting a performance of 56% compared to 21% for the Ctrl-X method.

[*AC's thoughts on the response*]
The authors provide a strong rebuttal to address several of the reviewer’s concerns. However, two important issues appear to be only partially addressed or remain unresolved, namely the fundamentally empirical nature of the approach and its limited generality across diffusion architectures. As such, it is unlikely that the reviewer would have increased the score.

**Reviewer ogmb:** Authors reiterate their response regarding the computational study detailing the overhead introduced by the RR and ARP modules and acknowledge that extending the method to DiT-style architectures would require additional effort.

---

### Decision · Program_Chairs · 2026-01-26

Reject